# Cancer-associated Histone H3 N-terminal arginine mutations disrupt PRC2 activity and impair differentiation

Benjamin A. Nacev [1,2,3] ✉, Yakshi Dabas[4], Matthew R. Paul[5], Christian Pacheco[4], Michelle Mitchener[6], Yekaterina Perez [7], Yan Fang[4,8], Alexey A. Soshnev [4,12], Douglas Barrows[5], Thomas Carroll[5], Nicholas D. Socci[9], Samantha C. St. Jean [10], Sagarika Tiwari[1,3], Michael J. Gruss[1,3], Sebastien Monette [10], William D. Tap[8], Benjamin A. Garcia [11], Tom Muir [6] & C. David Allis [4,13]

Dysregulated epigenetic states are a hallmark of cancer and often arise from genetic alterations in epigenetic regulators. This includes missense mutations in histones, which, together with associated DNA, form nucleosome core particles. However, the oncogenic mechanisms of most histone mutations are unknown. Here, we demonstrate that cancer-associated histone mutations at arginines in the histone H3 N-terminal tail disrupt repressive chromatin domains, alter gene regulation, and dysregulate differentiation. We find that histone H3R2C and R26C mutants reduce transcriptionally repressive H3K27me3. While H3K27me3 depletion in cells expressing these mutants is exclusively observed on the minor fraction of histone tails harboring the mutations, the same mutants recurrently disrupt broad H3K27me3 domains in the chromatin context, including near developmentally regulated promoters. H3K27me3 loss leads to de-repression of differentiation pathways, with concordant effects between H3R2 and H3R26 mutants despite different proximity to the PRC2 substrate, H3K27. Functionally, H3R26C-expressing mesenchymal progenitor cells and murine embryonic stem cell-derived teratomas demonstrate impaired differentiation. Collectively, these data show that cancer-associated H3 N-terminal arginine mutations reduce PRC2 activity and disrupt chromatin-dependent developmental functions, a cancer-relevant phenotype.

Histone proteins, together with associated DNA, form nucleosomes, the basic subunit of chromatin. Combinatorial post-translational modifications (PTMs) of histones regulate critical cellular processes such as cell fate, DNA damage repair, and the cell cycle[1]. Not surprisingly, genetic alterations affecting the machinery that deposits, recognizes, and removes histone PTMs (i.e., 'writers', 'readers', and 'erasers') can function as oncogenic drivers and are of intense interest in anticancer drug development[2–5].

In addition, dominant cancer-associated missense mutations in histones themselves (termed 'oncohistones') were initially discovered at three amino acids spanning a nine amino acid region of the histone H3 N-terminal tail. These include mutations at lysine 36 (H3K36M) in chondroblastoma, chondrosarcoma, undifferentiated pleomorphic sarcoma, and head and neck squamous cell carcinoma; at glycine 34 (H3G34W/L/R) in giant cell tumors of the bone, osteosarcoma, and glioma; and at lysine 27 (H3K27M) in diffuse intrinsic pontine glioma[6–11]. These "classical" oncohistones inhibit the activity of H3K27 or H3K36 methyltransferases leading to aberrant chromatin landscapes and dysregulated chromatin-dependent functions including differentiation and DNA damage repair[8,12,13].

In contrast to most proteins, which are encoded by a single gene, more than a dozen genes encode identical or highly similar histone H3 proteins. In the case of classical oncohistones, a missense mutation is observed in a single histone encoding gene. Thus, the capacity for an oncohistone to exert a phenotype in the context of a mostly wild-type histone H3 protein pool is critical for oncohistone-induced chromatin changes. The lysine to methionine (K-to-M) subset of classical oncohistone mutations inhibit their cognate methyltransferases with marked effects in trans (i.e., on wildtype H3) characterized by widespread loss of histone methylation products. In contrast, mutations at H3G34 function biochemically *in cis* with effects restricted to the histone tail harboring the mutation[13]. This *in cis* activity is explained by the requirement for the glycine 34 residue in the minus 2 position relative to the lysine substrate for H3K36 methyltransferases[14].

Despite the restricted direct biochemical sequelae of H3G34 expression, H3G34 mutants nonetheless induce locus-specific chromatin reorganization through loss of H3K36me3 at specific domains[15,16]. Such changes alter gene regulation and drive cancer phenotypes[15]. The extension of histone PTM changes from an *in cis* biochemical effect to domain reorganization are hypothesized to result from the polymeric nature of chromatin wherein incorporation of an abnormal histone subunit at low frequency can serve as a nucleation event for the propagation of effects through the polymer[8,17].

We and others have observed recurrent cancer-associated histone missense mutations outside the nine amino acid stretch of the H3 tail where classical oncohistones are found[18–20]. This expanded set of mutations occur in all four core histones at hotspots in both the tail and structured globular domains. Subsets of these mutations are now understood to affect the activity of chromatin remodelers and chaperones and, like the classical oncohistones, can perturb gene expression and differentiation[19,21]. However, in most cases the mechanisms and consequences of these additional oncohistone mutations are poorly understood[22].

Here we show that a subset of oncohistone mutations that occur at arginine residues in the H3 N-terminal tail and act *in cis* to alter the deposition of key histone posttranslational modifications, perturb H3K27me3 domains through disruption of PRC2 function, and promote aberrant differentiation programs that affect lineage commitment in progenitor populations. These findings reveal effects of H3 N-terminal tail arginine mutations in cancer-relevant biologic

functions, which may at least partially explain why they are recurrently observed in tumors.

## Results

### H3 N-terminal arginines are recurrently mutated in cancer

Our prior analysis of 4205 cancer-associated somatic histone missense mutations identified recurrent mutations of arginine residues in the N-terminal tail of histone H3.1 (hereafter H3)[18]. There are four arginines in this region, three of which—R2, R8, and R26—are mutated with frequencies similar to or exceeding those of classical oncohistones at H3K27, G34, and K36 (Fig. 1A) and are observed in a diverse group of carcinomas and sarcomas (Supplementary Fig. S1, Supplementary Data 1). In addition, when compared to the mutation count of nearby amino acids (Fig. 1B), mutation frequencies at the arginine residues exceed the local background. This is particularly the case for H3R26, which is the 7th most frequently mutated amino acid in the entire histone mutation cohort, independent of core histone family and tail/histone fold domain association[18]. The fourth arginine in the H3 N-terminal tail, H3R17, was not frequently mutated compared to other amino acid positions in this region (Fig. 1B). Notably, H3R2, 8, and R26 mutations were observed in a cohort of tumors with infrequent mutations (tumor mutation burden ≤2 mutations/Mb)[18], suggesting potential positive selection and implied functionality.

### H3 N-terminal R2 and R26 mutations impair PRC2 activity

Given the relative prevalence of H3R2, R8, and R26 mutations compared to classical oncohistones and the proximity of these amino acids to H3 N-terminal lysine PTM sites, we sought to test the hypothesis that H3-arginine cancer-associated missense mutations alter histone PTM prevalence. In tumors, a variety of missense mutations occur at the H3 arginine residues of interest. We therefore selected a commonly observed arginine to cysteine substitution to model (Supplementary Data 1). To do so, we expressed C-terminal HA- and FLAG-tagged transgenic H3 arginine mutants or a wild-type H3 (H3WT) control in HEK293T cells to measure their effects on PTM levels using PTM-specific antibodies. This system has previously been used to study the effects of classical oncohistones[8,13,16].

The prevalence of H3 methylation at different H3 N-terminal lysines was reduced in a manner dependent on the presence and

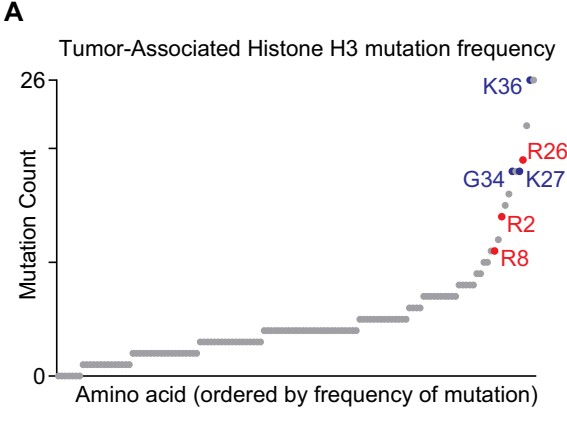

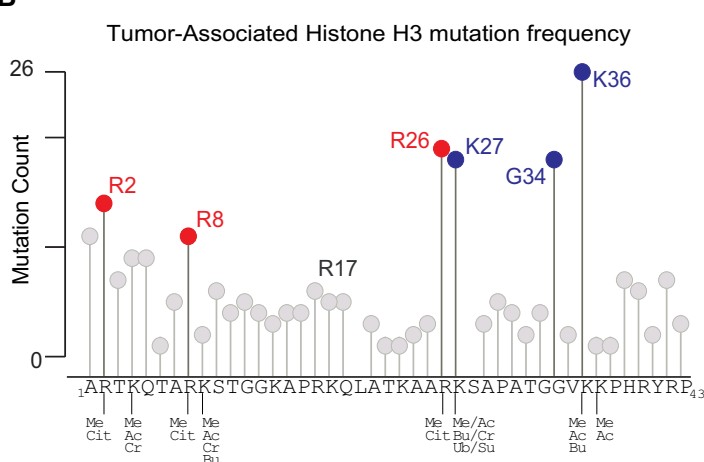

**Fig. 1 | Histone H3 N-terminal arginine mutations are prevalent in cancer.** Using data from a published dataset of cancer-associated histone mutations[18], **A** the frequency of mutations (y-axis) in histone H3 is plotted for each amino acid in the protein (x-axis) ranked by increasing frequency of mutations or **B** by sequence position with the single letter amino acid sequence of H3.1 noted (only positions 1–43 are shown). Blue indicates classical oncohistones. Red indicates H3 N-terminal arginine residues of interest. Selected sites of histone post-translational modifications[64] are denoted (Me methyl, Cit citrullination, Ac acetylation, Cr crotonylation, Bu butyrylation, Ub ubiquitination).

specific location of a H3R mutation when compared to wild-type H3 (H3WT) expressed at similar levels (Fig. 2A). Both H3R2C and H3R8C mutants led to decreased methylation at H3K4, with the most profound effects on H3K4me3, a PTM typically associated with transcriptionally active promoters. H3R8C markedly reduced the heterochromatin-associated histone modification H3K9me3, and H3R26C resulted in near complete loss of H3K27me3, which promotes

transcriptional silencing. Expression of H3R26C had similar effects on H3K27me1/2 and led to an expected reciprocal increase in H3K27ac. H3R2C also decreased H3K27me3 with less pronounced effects on H3K27me1/2, which was unexpected given its distance from H3K27 in the sequence of the unstructured tail. Notably, a mutation at H3R17, which is not frequently mutated in cancer, did not have any effects on PTM levels.

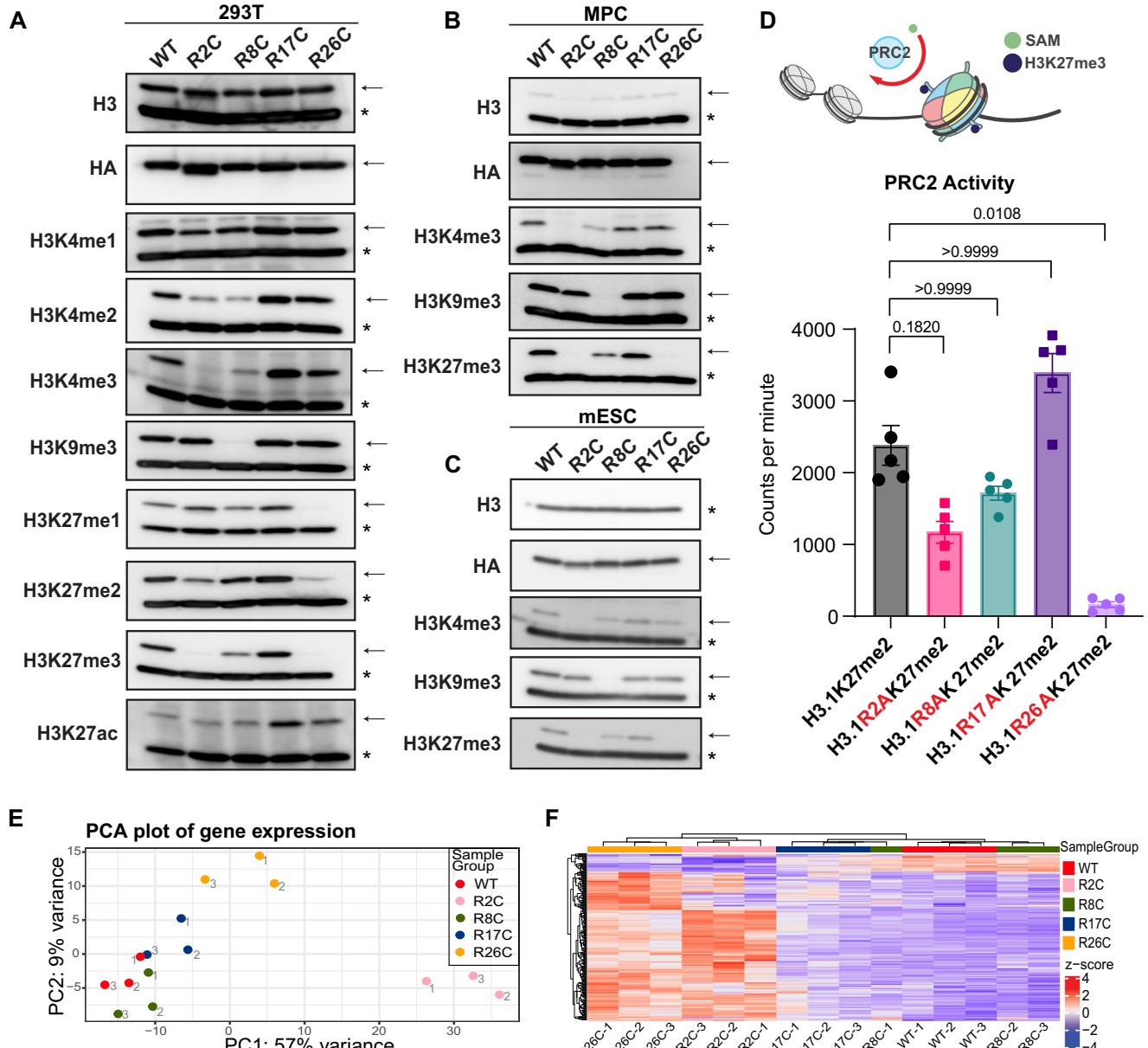

**Fig. 2 | H3R2 and H3R26 histone mutations similarly disrupt histone PTMs, PRC2 activity, and transcription. A** Immunoblot analysis of total H3, HA-epitope-tagged transgenic H3, or specific histone PTMs using PTM-specific antibodies in lysates from H3WT or H3 arginine mutant expressing HEK293T (293T), **B** mesenchymal progenitor cells (MPC), or **C** murine embryonic stem cells (mESC). The epitope-tagged mutant (or wildtype control) histone is marked by an arrow, whereas the endogenous wildtype H3 histone is marked by an asterisk. The approximate molecular weight of endogenous H3 (~16 kDa) was determined by reference to a molecular weight ladder and the stereotypical core histone migration pattern by Direct Blue staining of uncropped membranes corresponding to each blot, which are available in the Source Data file along with uncropped images of the immunoblots. Blots shown in panels **A–C** are representative of duplicate experiments. **D** In vitro PRC2 methyltransferase activity for H3K27me2 nucleosome array substrates harboring H3WT versus mutant H3 using tritiated S-

adenosyl-L-methionine (SAM) as a methyl donor. PRC2 activity is shown as counts per minute after background (no enzyme) counts were subtracted. Group means were determined to significantly differ ($p < 0.001$) using a nonparametric test (Friedman) of matched replicates ($n = 5$ independent assays). Dunn's test (two-sided) was then used to compare each mutant to the H3WT control. $p$-values are shown. Error bars are standard error of the mean. Source data are provided as a Source Data file. **E** Principal component analysis of bulk RNA-seq from MPC expressing H3WT or H3R2C, H3R8C, H3R17C, or H3R26C. Three individual biologic replicates are shown for each genotype. The point labels indicate the replicate number. **F** H3WT or H3 arginine mutant MPC clustered based on expression values of differentially expressed genes in H3R26C vs H3WT. Only genes with at least a 2-fold change in expression and $p$-adj < 0.05 are shown. Three independent replicates are shown, each individually denoted by the suffix "-n".

Similar PTM changes were observed when the H3R mutants were expressed in two additional cell lines, C3H/10T1/2 murine mesenchymal progenitor cells (MPC) and murine embryonic stem cells (mESC) (Fig. 2B, C). Despite lower expression of the transgene relative to the total histone H3 pool in MPC compared to HEK293T, and an even lower expression of the transgene in mESC, mutant effects on PTMs were analogous. Importantly, H3WT was expressed at similar levels to the H3R mutants in each of the three cell lines. In all cell lines, the observed changes in PTMs occurred *in cis*, with loss of the PTMs on the transgenic histone but without appreciable changes in the endogenous histone pool. In addition, the expression of core PRC2 subunits was not reduced by H3R26C expression relative to H3WT in either MPCs or HEK293T cells. While PRC2 subunit expression was reduced in the H3R2C mutant relative to H3WT in MPCs cells, it was not in HEK293T. Thus, changes in PRC2 subunit expression do not explain the effects of H3R2C or H3R26C on H3K27 methylation (Supplementary Fig. S2A).

The nuclear localization of the transgenic histones was confirmed by immunofluorescence (anti-HA) with chromatin association supported by colocalization of the anti-HA signal with condensed mitotic chromatin (Supplementary Fig. S2B). Since the arginine-to-cysteine substitutions introduce a reactive sulfhydryl group, which could lead to protein crosslinking and represent a mechanism for PTM changes distinct from other missense mutations at the same position, we alternatively substituted alanine or histidine at each of the arginines of interest. In all cases, we observed similar effects on H3K4me3 and H3K27me3 in HEK293T cells when compared to R-to-C mutations (Supplementary Fig. S2C). Given the proximity of H3R26 to H3K27, to ensure that H3K27 epitope loss or occlusion by the H3R26 mutants was not confounding the immunoblot results, we performed middle-down mass spectrometry, which confirmed a near total *in cis* loss of H3K27me3 in H3R26C mutant histone tails (Supplementary Fig. S2D, Supplementary Data 2). Finally, to ensure that H3K27me3 could be detected by a PTM-specific antibody in the absence of the wild-type H3R26 amino acid, we confirmed that recombinant H3R26A-K27me3 and H3WT-K27me3 nucleosomes were similarly recognized by the anti-H3K27me3 antibody (Supplementary Fig. S2E).

Notably, both H3R2 and H3R26 can be methylated raising the possibility that loss of these methylation sites in the mutant could contribute to decreased lysine PTMs via crosstalk mechanisms. However, prior studies have established that that H3R2me2 is anti-correlated with H3K4me3 and enriched at inactive promoters[23] and that H3R2me2s impairs PRC2 activity compared to unmethylated H3R2 substrates[24]. To test these relationships in our system, we treated MPCs with a type I arginine methyltransferases (MS023)[25], an inhibitor of the predominant type II arginine methyltransferase, PRMT5 (GSK591)[26], or both, to deplete methylarginine. Type I and type II arginine methyltransferases catalyze monomethylarginine (MMA), type I arginine methyltransferases catalyze asymmetric dimethyl arginine (ADMA), and type II arginine methyltransferases catalyze symmetric dimethyl arginine (SDMA)[27]. We established doses of each drug that reduced MMA, ADMA, and SDMA using PTM-specific antibodies applied to whole cell extracts (Supplementary Fig. S3A). SDMA was detected in the core histone range in acid extracted histones and was reduced upon treatment with the type II arginine methyltransferase inhibitor (GSK591 alone or with MS023) suggesting that these doses were also sufficient to reduce histone arginine methylation (Supplementary Fig. S3B). At the same doses of arginine methyltransferase inhibitors, there was no reduction in H3K4me3 or H3K27me3 on the transgenic H3WT tail or gain of either on the H3R2C mutant histone tail (Supplementary Fig. S3C). On the contrary, co-inhibition of type I and II arginine methyltransferases appeared to slightly increase H3K4me3 and H3K27me3, which is in keeping with prior work. In addition, PTM profiling using mass spectrometry did not detect any H3R2 methylation and H3R26me1/2 was detected in <10% of H3WT population (Supplementary Data 2), which is consistent with

prior studies[28]. Thus, loss of H3R2- and H3R26 methylation is unlikely to explain the observed effects on H3K4me3 and H3K27me3 loss.

## Histone H3R26 is required for PRC2 histone methyltransferase activity

The Polycomb Repressive Complex 2 (PRC2), the 'writer' of H3K27 mono-, di-, and tri-methylation, is a critical regulator of lineage commitment programs and is commonly disrupted in cancer[29]. One explanation for the observed decreased H3K27me3 in cells expressing H3R2C or R26C mutants is reduced PRC2 activity. To assess this, we determined the effect of all the H3 arginine mutants in an in vitro PRC2 methyltransferase assay in which PRC2 activity was measured by incorporation of a tritiated methyl group from an $^3$H-labeled S-adenosyl-L-methionine donor. The substrate in this assay was a 12-mer homogenous nucleosome array of recombinant nucleosome octamers generated with H3WT or H3 arginine mutants (Supplementary Figs. S2F, S2G). In all the H3R mutants, the arginine was substituted to alanine since this was more compatible with the synthetic process and the R-to-A missense mutations affected H3K27 methylation similarly to a cystine substitution when analyzed by immunoblot (Supplementary Fig. S2C). The recombinant histone H3 was dimethylated in all substrates to allow us to specifically assess the production of H3K27me3, since PRC2 generates H3K27me1/2/3.

Compared to a wild-type substrate, the PRC2-catalyzed methylation of K27-dimethylated H3R26A arrays was largely eliminated (6.8%, $p = 0.01$), which agreed with the immunoblot data (Fig. 2D). The methylation of H3R2A arrays was reduced (49%) though did not reach statistical significance ($p = 0.18$), suggesting that the complete loss of H3K27me3 on H2R2 mutant histones in the cellular context as determined by immunoblot is mediated by factors in addition to direct effects on the core PRC2 complex. H3R8 and R17 mutants, which were associated with minimal or no loss of H3K27me3 in immunoblot assays did not significantly differ from H3WT as PRC2 substrates in the in vitro methyltransferase assay ($p > 0.99$).

Because PRC2-catalyzed H3K27me3 has an important role in silencing gene expression and both H3R2A and H3R26A reduced PRC2 activity in vitro and/or in cells, we speculated whether these mutants could alter transcription compared to H3WT expressing cells and if the changes would be similar. To address this, we examined gene expression in H3WT, R2C, R8C, R17C, and R26C expressing MPC, a cell type which we previously used to study oncohistone-induced disruption of chromatin and chromatin-dependent gene regulation and differentiation[8,21]. The bulk transcriptomes of H3R2C and H3R26C MPC differed from H3WT and the other H3R mutants in a principal component analysis (Fig. 2E). Consistent with the silencing function of H3K27me3 and the near complete loss of in vitro PRC2 activity for a H3R26 mutant substrate, MPC expressing H3R26C significantly upregulated expression of 226 genes ($p$-adj < 0.05, $\log_2$FC > 1) (Supplementary Data 3). In contrast, only 28 genes were downregulated. The expression changes in these differentially expressed genes clustered with similar changes in the H3R2C mutant MPC but not with H3WT, H3R8C, or H3R17C mutant MPCs (Fig. 2F). Thus, H3R2 and H3R26 mutants had convergent effects of reduced H3K27me3 and transcriptional de-repression of similar genes.

## Histone H3R2 and R26 mutations perturb H3K27me3 chromatin domains

Having observed similar transcriptional changes in H3R2C and H3R26C compared to H3WT and other H3 arginine mutants, we wondered whether these mutations could broadly alter chromatin or if the effects were restricted to the mutated histone tail *in cis*. We therefore profiled H3K27me3 using CUT&RUN[30] and compared the H3K27me3 signal in H3R2C and R26C MPC to signal in H3WT cells in three replicate experiments, noting that the H3K27me3 antibody used can recognize H3K27me3 in the absence of H3R26 (Supplementary

Fig. S2E). In the mutants, we observed significant gains and losses of H3K27me3 signal at H3K27me3 peaks (Supplementary Data 4, Supplementary Fig. S4A), compatible with a locus-specific effect. Overall, the H3K27me3 signal at consensus peak regions was decreased in the mutants as was the median signal for all peaks with more notable effects at greater signal intensity (Supplementary Fig. S4B–F). The genes associated with the differential H3K27me3 peaks in H3R2C and R26C mutants, but not a randomly permuted gene cohort, corresponded to documented Polycomb-regulated gene sets (Fig. 3A, Supplementary Fig. S5).

Comparison of the H3K27me3 signal between H3WT, H3R2C, and H3R26C at differential H3K27me3 peaks in H3R2C or H3R26C revealed a similar trend of overall decreased signal intensity in the mutants (Fig. 3B, C). Close examination of H3K27me3 at specific regions of differential peaks further revealed subsets of peaks where H3K27me3 was lost in both mutants, only one or the other mutant, or, in some cases, gained in as an extension of an H3WT peak region (Fig. 3D, E), Supplementary Fig. S6A,B). The mutant-specific loss of H3K27me3 in H3R2C or H3R26C was most apparent in peaks with the greatest magnitude of H3K27me3 signal loss compared to H3WT (Fig. 3F, G).

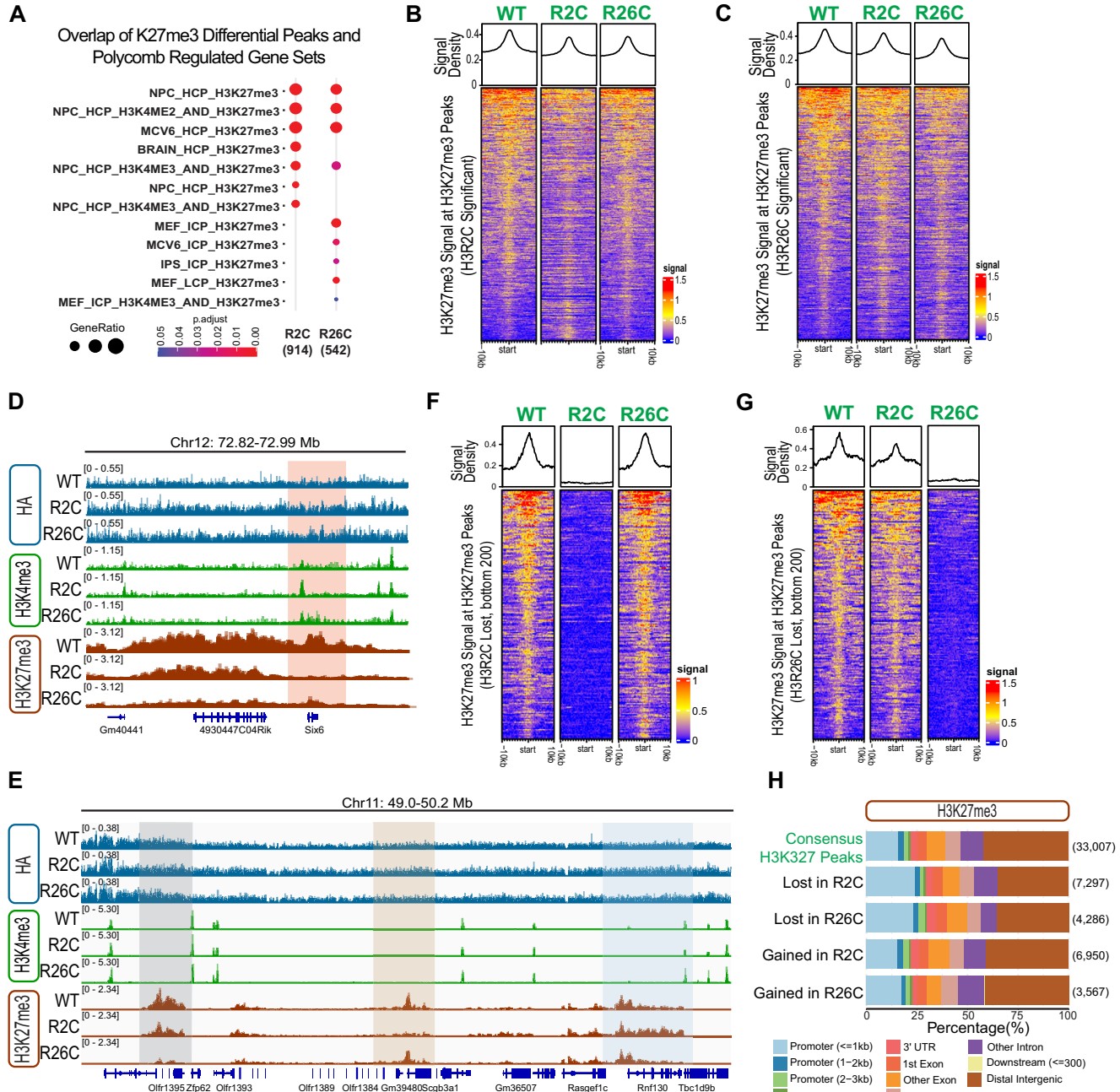

**Fig. 3 | H3R2 and H3R26 mutants alter H3K27me3 in a locus-specific manner.**
**A** Enrichment of differential H3K27me3 peaks in known polycomb-regulated datasets. **B** H3K27me3 signal at H3K27me3 differential peaks in H3R2C or **C** H3R26C. **D** IGV (version 2.15.4) tracks of representative regions, with differential H3K27me3 peaks (mutant/WT) lost in both H3R2C and HR26C (red shading), **E** lost in H3R2C alone (beige shading), H3R26C alone (gray shading) or gained in H3R2C (blue shading). Histograms are overlayed for 3 independent replicates for each genotype (see Supplementary Fig. S3A, B for separated individual replicates). **F** H3K27me3 differential peaks (H3R2C versus H3WT) with the greatest magnitude loss in H3R2C (bottom 200 peaks). **G** H3K27me3 differential peaks (H3R26C versus H3WT) with the greatest magnitude loss in H3R26C (bottom 200 peaks). **H** Types of genomic regions associated with lost H3K27me3 peaks in H3R2C and H3R26C compared to consensus peaks. The number of peaks in each group is indicated in parentheses.

When all mutant-specific differential peaks are taken into account, there was more similarity between H3R2C and R26C samples at H3R2C-lost H3K27me3 peaks than at H3R26C-lost peaks based on overall H3K27me3 signal density (Fig. 3B, C).

When comparing the chromatin incorporation of the transgenic H3R2C and R26C with H3WT using CUT&RUN for the HA-epitope tag, there was no enrichment of the mutants compared to H3WT in the regions of interest (Fig. 3D, E, Supplementary Fig. S6A, B). Analysis of the HA signal at all H3K27me3 differential peaks did not reveal any significant differences in HA signal of more than one $log_2$-fold change between H3R26C and H3WT and fewer than ten such peaks in H3R2C (Supplementary Data 5, Supplementary Fig. S4F). This was markedly different from the magnitude and number of highly significant differences in H3K27me3 signal at H3K27me3 differential peaks in the same samples (Supplementary Fig. S4A), strongly suggesting that hypothetical locus-specific differences in histone mutant incorporation density do not explain the observed changes in broad regions of H3K27me3.

Because H3R2C and R26C mutants reduced PRC2 activity and H3K27me3 differential peaks in these mutants overlapped with Polycomb-regulated genes, we hypothesized that the lost H3K27me3 peaks were associated with genomic features important for gene regulation. Compared to consensus peaks, loss of H3K27me3 occurred more often within 1 kb of promoters, and less frequently at distal intergenic regions (Fig. 3H). In contrast, peaks gained in H3R2C or R26C were not enriched at specific features. Thus, H3K27me3 was lost in regions where PRC2 has a transcriptionally repressive function.

Since the H3R2C mutant reduced both H3K4me3 and H3K27me3 on the mutant histone tail (Fig. 2A–C), we explored the overlap of these PTMs at specific loci using CUT&CUN in the H3R2C- versus H3WT-expressing MPCs (Supplementary Data 6). Increased H3K27me3 peaks significantly overlapped with decreased H3K4me3 peaks (enrichment score 10.39, $p < 1.38 \times 10^{-47}$), which is expected given that in general, H3K4me3 and H3K27me3 mark different chromatin states[1]. We also observed a second significantly overlapping set of peaks corresponding to regions where H3K4me3 and H3K27me3 were both decreased in the H3R2C mutant expressing MPCs compared to wildtype (enrichment score 8.00, $p < 2.30 \times 10^{-58}$). There was no significant overlap of regions that lost H3K27me3 and gained H3K4me3 (enrichment score 1.08, $p < 0.35$).

The regions where both H3K4me3 and H3K27me3 were lost may reflect loci directly affected by the H3R2C mutant given concordance with the observed loss of both H3K4me3 and H3K27me3 on the H3R2C mutant histone (Fig. 2A–C). Further supporting a potential direct effect, the genes associated with overlapping decreased H3K4me3 and H3K27me3 peaks in the H3R2C mutant are enriched pathways associated with polycomb regulated genes (Supplementary Fig. S7A, Supplementary Data 6). In contrast, the association between gained H3K27me3 and lost H3K4me3 is more likely indirect, perhaps through redistribution of H3K27me3. From a standpoint of potential function, genes at overlapping decreased H3K4me3 and H3K27me3 peaks in the H3R2C mutant are significantly associated with GO terms involved in differentiation and cell fate including "mesenchymal differentiation", "stem cell differentiation", and "muscle organ development" (Supplementary Fig. S7B, Supplementary Data 6).

### Histone H3R2 and R26 mutations disrupt transcriptional control of growth and differentiation pathways

Following from the observation of locus specific H3K27me3 loss at polycomb regulated genes and the transcriptional activation of similar genes in H3R2C and H3R26C mutants, we next determined if these genes were associated with specific programs. This would be consistent with a functional effect on PRC2-mediated regulation of development and lineage commitment. Gene ontology (GO) analysis of differentially expressed genes in H3R26C versus H3WT MPC

revealed an enrichment in pathways related to differentiation including blood vessel morphogenesis, regulation of vascular development, endothelial cell migration, and extracellular matrix organization (Fig. 4A). Of the 1316 differentially expressed genes (*p-adj* < 0.05, | $log_2FC|>0$) that overlapped in H3R2C and H3R26C, nearly all (97%) changed in the same direction relative to H3WT (Supplementary Data 7), suggesting a convergent mechanism of transcriptional dysregulation. GO analysis of this overlapping gene set yielded terms specifically associated with mesenchymal lineages, including osteoblast differentiation (e.g., *Bmp4, Sox9*), regulation of fat cell differentiation (e.g., *Cebpa, Pparg*), regulation of smooth muscle proliferation (e.g., *Notch3, Pdgfrb*), and angiogenesis (e.g., *Tgfb2, Nfatc4*) (Fig. 4B, Supplementary Data 8). Thus, in a mesenchymal progenitor context, both H3R26C and R2C mutants affected overlapping sets of lineage-relevant and tightly regulated developmental programs.

Three lines of evidence suggest that these transcriptional effects are driven by the loss of PRC2 activity. First, the H3R2C and R26C mutants, which reduce PRC2 activity, similarly alter the transcriptome of MPC compared to H3WT expressing MPC, whereas MPC expressing H3R8C and R17C, which do not reduce PRC2 activity, have transcriptomes similar to H3WT. Second, the majority of differentially expressed genes were upregulated, consistent with the widely accepted role of PRC2 as a transcriptional repressor. Third, affected gene sets were related to lineage commitment or differentiation programs, which are regulated by PRC2.

To understand how H3K27me3 changes correlated with transcription of nearby genes, we examined gene expression in regions where H3K27me3 was gained, lost, or did not significantly change in H3R26C versus H3WT MPC. Genes associated with H3K27me3 loss were increasingly transcribed compared to genes without change in H3K27me3 ($p = 0.0041$) and those that gained H3K27me3 ($p = 2.9 \times 10^{-5}$) (Fig. 4C). Gene expression in regions of H3K27me3 gain was not significantly different than at genes where there was no change in H3K27me3 ($p = 0.14$).

We next determined DNA sequence motifs where H3K27me3 was specifically lost in H3R26C-expressing MPC and compared these to transcription factor binding sequences. This revealed thirty transcription factors that putatively interact with these sites (Supplementary Data 9). Many of these bind an E-box motif (5′-CANNTG-3′) and include members of the helix-loop-helix (bHLH) family of transcription factors[31], which were highly represented, particularly among the top ranked sequences. Notably, bHLH transcription factors regulate development and lineage programs and two bHLH family transcription factors important for muscle development, MYOD1 and MYOG,[32] were identified as putative binders of the H3K27me3-depleted sites (Fig. 4D). Consistent with potential functional modulation of this pathway upon loss of a repressive mark, gene set enrichment analysis revealed a positive correlation between H3R26C expression and expression of a myogenesis gene set (Fig. 4E). Thus, H3R26C expression was additionally linked with dysregulation of pathways relevant for mesenchymal lineage development.

### H3R26C mutations perturb mesenchymal differentiation in MPC

Given the significant effect of H3R26C on gene expression programs related to mesenchymal development and the role of PRC2 therein[33], we postulated that mesenchymal differentiation would be influenced by H3R26C expression. We tested this using a model in which MPC stochastically differentiate into multiple mesenchymal lineages, including adipocytes and skeletal muscle, following a brief treatment with the DNA methyltransferase inhibitor, 5-Azacitidine (5-Aza)[34]. Focusing first on adipogenesis, we observed robust differentiation of H3WT-expressing MPCs to adipocytes as determined by staining with the fluorophore-conjugated lipid dye, LipidTOX, which appeared reduced in H3R26C-expressing MPC (Fig. 5A). To measure this

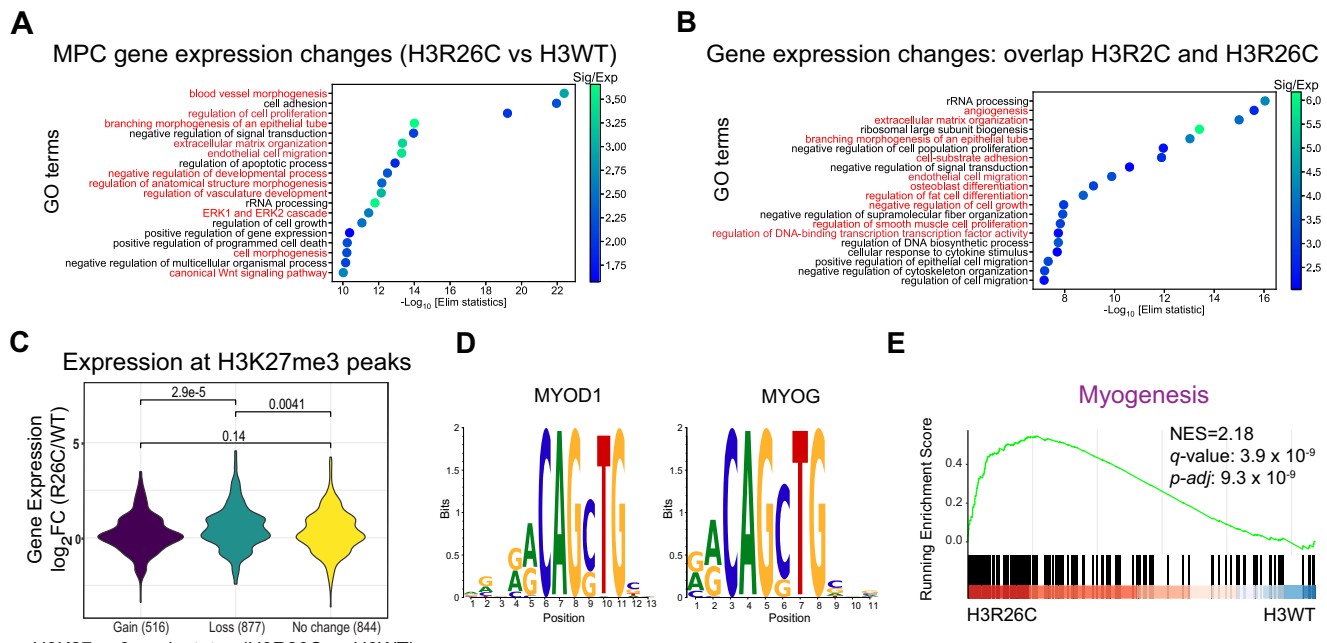

**Fig. 4 | H3R2C and H3R26C histone mutants disrupt transcriptional regulation of developmental and growth pathways.** **A** Gene Ontology analysis (TopGo) of differentially expressed genes when comparing H3R26C to H3WT or **B** genes differentially expressed in both H3R2C and H3R26C when each were compared individually to H3WT and when the direction of change was the same in both mutants. Red lettering indicates pathways relevant to linage specification or differentiation. **C** Change in gene expression at H3K27me3 associated genes based on H3K27me3 differential peak status (gain, loss, or no change; number of peaks denoted in parentheses) in H3R26C versus H3WT. p-values are shown above the brackets for the pairwise comparison between groups based on a two-sided Wilcoxon test. **D** Motif-analysis of sequences associated with H3K27me3 peak loss revealed putative MYOD1 and MYOG consensus sequences using the CIS-BP 2.00 database. **E** Gene set enrichment analysis of H3R26C vs H3WT demonstrated enrichment for genes in the myogenesis pathway from MSigDB Hallmarks (NES = 2.18, $q$-value = 3.9 ×10$^{-9}$).

difference, we calculated an adipogenic index (AI)[35], determined by the percent of total area with LipidTOX staining per total nuclei count (Supplementary Fig. S8), which was significantly reduced in H3R26C mutant expressing MPC compared to those expressing H3WT (two-tailed ratio paired t test; t = 5.289, df = 3, $p$ = 0.0132, 95% CI (0.1550, 0.6289), $R^2$ = 0.9032) (Fig. 5C). The mean ratio of the adipogenic indices (mutant/WT) for four replicates was = 0.335.

A similar effect of impaired differentiation was observed in another mesenchymal lineage, skeletal muscle. H3WT-expressing MPC formed multinucleated myotubes positive for the myocyte marker, Myosin 4, whereas H3R26C-expressing MPC expressed Myosin 4 in some cells, but were significantly restricted in their ability to form Myosin 4 positive mature multinucleated myotubes (two-tailed Mann–Whitney test, $p$ = 0.0286) (Fig. 5B, D).

### Expression of H3R26C alters germ layer equilibrium in embryonic stem cell differentiation

We next sought to determine if the effect of H3R26C on PRC2 mediated differentiation extended to an in vivo setting and beyond the mesenchymal context. We focused on mESC, since PRC2 is known to play a key role in lineage commitment therein[29]. This system allowed us to leverage the property of murine embryonic stem cells to form teratomas when grown as allografts. We have previously shown that disruption of epigenetic regulation in this context is sufficient to skew the relative proportion of germ layers within mESC-derived teratomas[36].

Therefore, we expressed H3WT and H3R26C in mESC where we had noted histone PTM changes, including the loss of H3K27me3 in the H3R26C mutant, were consistent with our observations in HEK293T and MPC (Fig. 2C). Following subcutaneous injection of H3R26C and H3WT mESC into host mice, teratomas formed in both genotypes that were competent to generate all three germ layers as well as an undifferentiated population (Fig. 5E). However, the relative abundance of the

germ layers was altered between H3R26C and H3WT, with a significant increase in ectoderm and a decrease in mesoderm in the H3R26C-expressing mESC (Fig. 5F). These data support an important consequence of H3R26C on tightly controlled developmental and lineage commitment programs in non-mesenchymal restricted contexts. This suggests a potentially important contribution of H3R26C and possibly other H3 arginine oncohistone mutations in multiple tumor types.

### H3R26C-mediated chromatin disruption is associated with aberrant or partially differentiated states

Based on the observation that H3R26C-expressing MPC differentiate to the extent that they express the Myosin 4 muscle marker but form mature myotubes less frequently than H3WT, we explored the possibility that H3R26C-expressing cells might aberrantly or only partially differentiate. After 5-Aza-induced differentiation, we used single cell RNA sequencing (scRNA-seq) to identify unique cell populations arising after cells received a differentiation cue. Unlike scRNA-seq of tissues from an intact organism where defined populations of cells are expected, we anticipated analyzing a mixture of undifferentiated progenitor cells, differentiated mesenchymal populations, and intermediate or partially differentiated cells. Therefore, we expected to observe only a few defined cell populations based on markers from mature mouse tissues. Among these, we identified a Ly6a+ cluster representing a mesenchymal precursor pool as well as clusters identified as myocytes and adipocytes based on their respective established markers (Fig. 6A, B, Supplementary Fig. S9A–C). Notably, in the adipocyte cluster, the ratio of 5-Aza-treated H3R26 C:H3WT was 0.386 (Fig. 6C), which is consistent with the ratio of H3R26C to H3WT adipogenic indices (0.335) as determined in independent experiments using an alternative method for adipocyte quantification (Fig. 5C).

To characterize additional populations, we annotated groups based on mesenchymal progenitor and lineage commitment

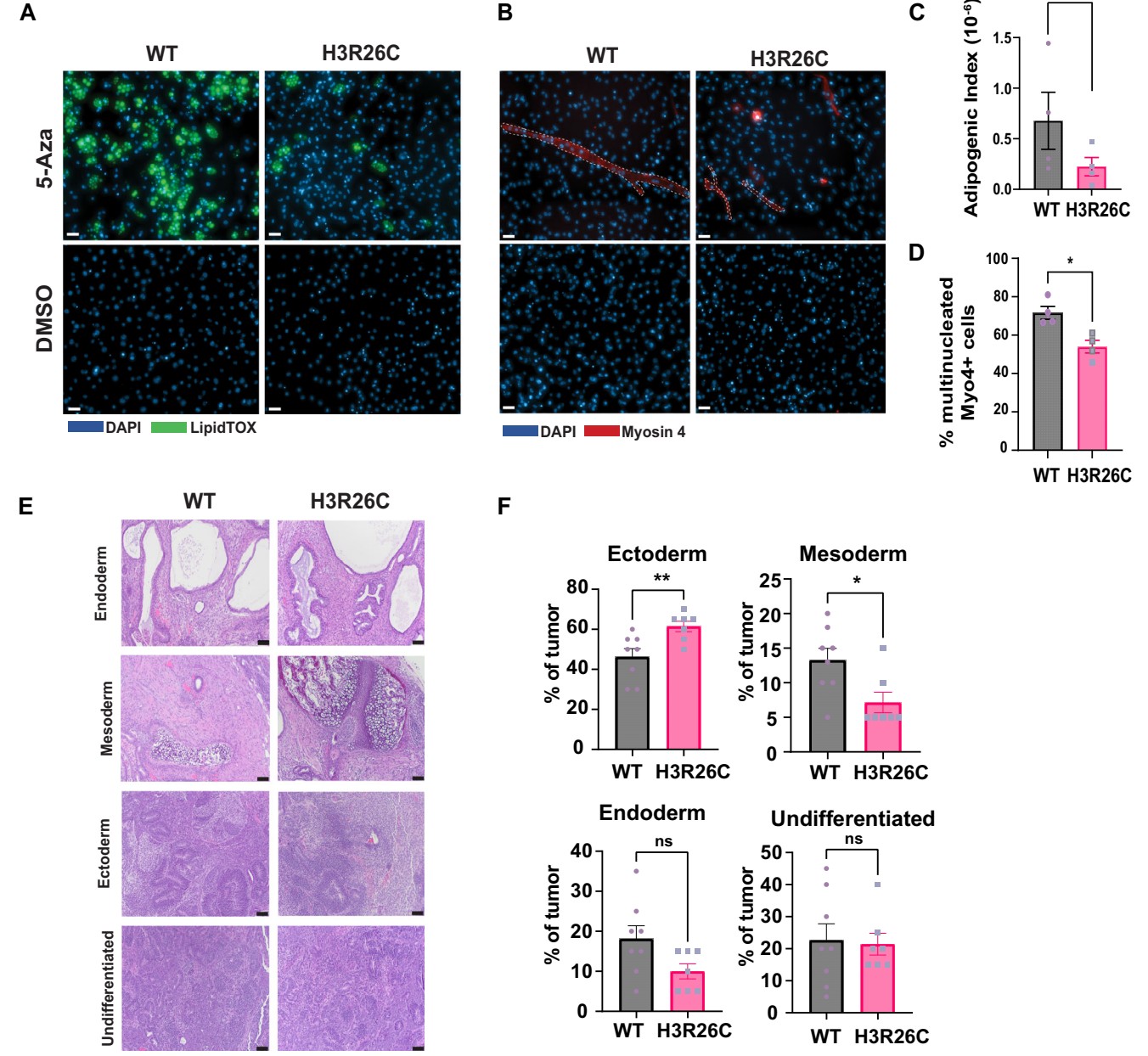

**Fig. 5 | Multilineage differentiation is altered by H3R26C expression.** Representative immunofluorescence images (Maximum Intensity Projections, MIPs, with brightness adjustment for visualization) of H3WT and H3R26C cells after treatment with 5-Azacitidine (5-Aza) or vehicle (DMSO) and staining with DAPI, **A** LipidTOX (adipocyte marker), or **B** Myosin 4 (Myo4, muscle linage marker). Bars = 50 μm. Myo4+ cells are outlined with white dashes. Results shown in panels **A** and **B** are representative of experiments repeated four times and quantified as in panels **C** and **D**. **C** Quantification of adipocyte differentiation using the Adipogenic Index ($n$ = 4 biologic replicates; Error bars are standard error of the mean; two-tailed ratio paired t test: t = 5.289, df = 3, *$p$ = 0.0132, 95% CI (0.1550, 0.6289), $R^2$ = 0.9032). Individual datapoints are shown. Source data are provided as a Source Data file. **D** Percentage of multinucleated Myo4+ cells ($n$ = 4 biologic replicates; Error bars are standard error of the mean; Two-tailed Mann–Whitney test, *$p$ = 0.028).

Individual datapoints are shown. Source data are provided as a Source Data file. **E** Representative micrographs from H3WT- and H3R26C-mESC-derived teratomas demonstrating all three germ layers and undifferentiated tissue. Bars = 100 μm. **F** Pairwise comparison of individual germ layer components and undifferentiated tissue in H3WT versus H3R26C. Individual datapoints are shown representing individual tumors grown in mice at bilateral flank injection sites ($n$ = 8 for H3WT and $n$ = 7 for H3R26C). For large tumors that required embedding in multiple blocks, the data from all blocks was averaged. An unpaired two-tailed t-test was used to compare groups; Ectoderm: t = 3.02, df=13, 95% CI(4.333, 26.02), $R^2$ = 0.4129, $p$ = 0.009; Mesoderm: t = 2.65, df = 13, 95% CI (−11.07, −1.142), $R^2$ = 0.3520, $p$ = 0.019; ns = $p$ ≥ 0.05. Error bars are standard error of the mean. Source data are provided as a Source Data file.

markers. We identified clusters (groups 2, 5, 8, 9) that expressed both the progenitor marker Ly6a and the early mesenchymal lineage marker Spp1 (Fig. 6B). The Spp1+ cluster is an aggregate of groups that expressed stemness and early lineage commitment markers. For example, group 2 expressed the adipogenic marker Plac8, group 9 expressed the stem cell markers CD34 and Hmg2a as well as Plac8, and group 8 expressed the stem marker Pdgfra together with Plac8 and second adipogenic marker, Enpp2. Notably, there was correlation between the bulk RNA-seq and scRNA-seq datasets. Transcripts representing significant GO terms from the bulk RNA-seq data (Fig. 4A) were enriched in specific clusters in H3R26C compared to the H3WT wildtype DMSO samples,

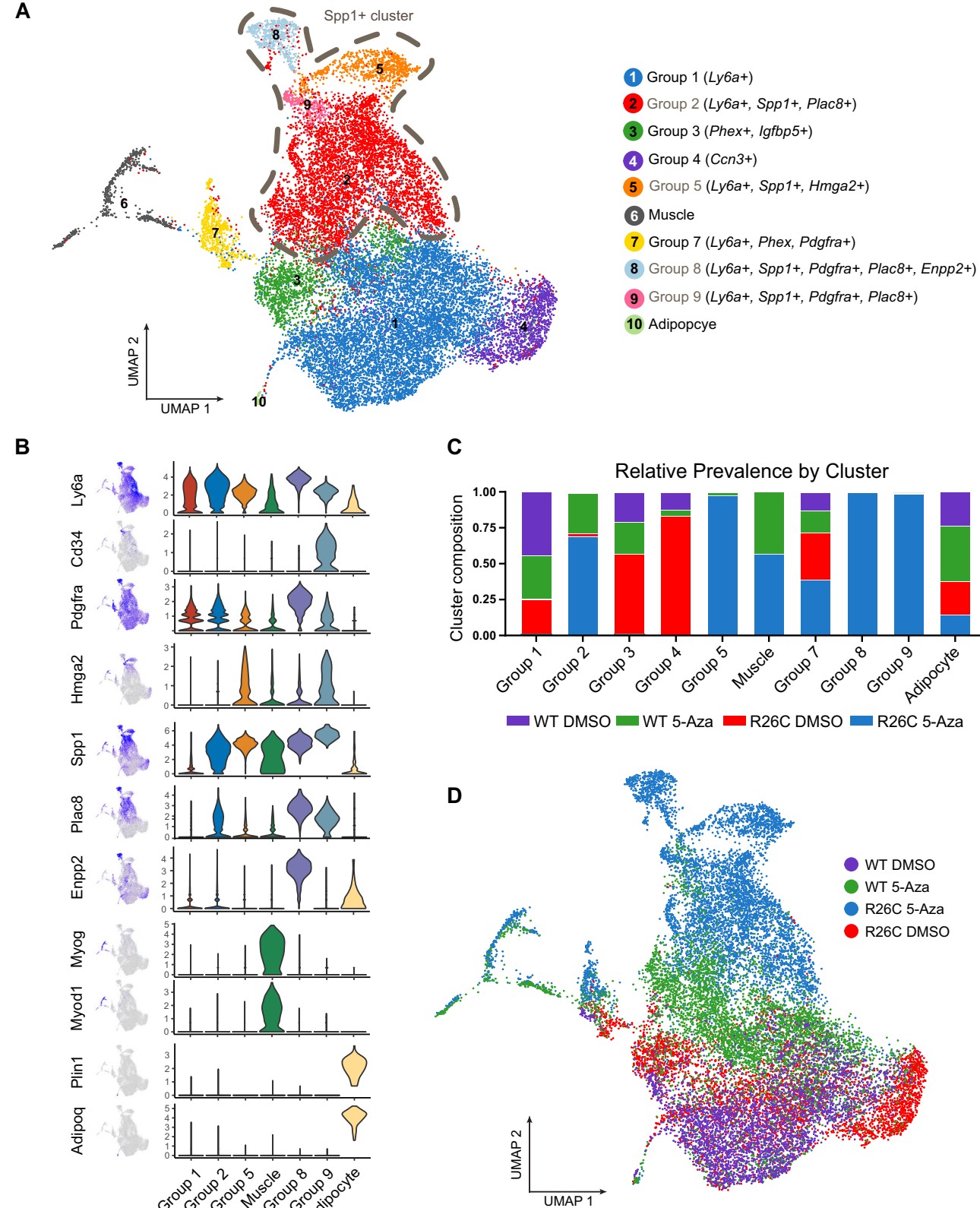

**Fig. 6 | H3R26C mutant expressing MPC form populations after differentiation induction that are distinct from H3WT.** H3WT and H3R26C expressing MPC were induced to differentiate with 5-Azacitidine (5-Aza) and transcriptionally profiled as single cells along with vehicle (DMSO) controls. **A** The transcriptomes of all four conditions are represented as ten clusters on a UMAP projection with relevant marker genes or linage assignments listed with the group names. The dashed line indicates Spp1+ groups, which are labeled with grey font. **B** The expression of individual marker genes was plotted for groups of interest and overlayed on the UMAP. **C** The proportion of sample (by genotype and treatment) represented in each group (source data are provided as a Source Data file) are plotted. **D** The sample identities are projected on the same UMAP as panel **A**.

particularly in Group 7 (Supplementary Fig. S9D–G). These differences persisted in the samples that received differentiation cues.

Having defined distinct populations, the contribution of each sample, defined by treatment conditions and genotype, was determined for each cluster (Fig. 6C, D). The Spp1+ groups, representing a hybrid of stemness and early markers of lineage commitment, were almost entirely composed of H3R26C cells that received a 5-Aza differentiation cue (Fig. 6C, D). By contrast, the groups with the greatest proportion 5-Aza-treated H3WT cells were muscle and adipocytes (Fig. 6C). Trajectory analysis revealed that the populations enriched for H3R26C cells that had received a differentiation cue (5-Aza) were more distant in pseudotime than vehicle treated cells or the WT 5-Aza-treated population (Fig. 7). As expected, the differentiated muscle cluster was also distant in pseudotime compared to the undifferentiated control clusters. Perhaps due to the relatively small population of cells, the adipocyte population was not resolved in the trajectory analysis. Collectively, these data support a model in which the H3R26C histone mutant disrupts differentiation by inducing an aberrantly differentiated population. This results in a reduction of the terminally differentiated population and the formation of unique populations only present in the mutant cells upon stimulation to exit the progenitor state.

## Discussion

Cancer-associated histone mutations occur throughout all four core histone families and in both N-terminal tail domains and structured globular domains. This work describes the functional consequences of a subset of oncohistone mutations observed at arginine residues in the N-terminal tail of histone H3.1. These H3 arginine mutations deplete key regulatory histone PTMs, including H3K4me3, H3K9me3, and H3K27me3. This effect is independent of histone mutant expression levels relative to the wildtype pool, as we observe comparable results in three different cell lines with a range of histone expression levels with matched mutant and H3WT control abundance in each.

The loss of H3K27me3 was particularly pronounced on H3R26 mutant histones and histone tails harboring this mutation served as a poor substrate for the PRC2 H3K27 methyltransferase. Interestingly, H3R2 mutants also reduced the PRC2 product, despite R2 being far removed from the K27 substrate in the unstructured histone tail. One possible explanation is that a core PRC2 subunit, Rbbp4, uses an acidic pocket to bind the H3 N-terminus[37], including the H3R2 sidechain. In the absence of H3R2, the binding of PRC2 to substrate nucleosomes may be diminished.

At the biochemical level, the H3 arginine mutants induced PTM changes, which were observed on the same tail harboring the histone mutant (i.e., *in cis*). This is reminiscent of the classical oncohistone mutations observed at H3.3G34 and in contrast to the classical oncohistone mutations that follow a K-to-M paradigm (e.g., H3.3K27M and H3.3K36M), which act in trans to diminish the global pool of H3K27me3 or H3K36me3, respectively. However, despite this *in cis* biochemical effect, remarkably both H3R2C and H3R26 mutants reduce H3K27me3 domains at specific loci in a pooled population of cells. Because an arginine in the −1 position of the $ARK_{27}S$ substrate motif is required for allosteric activation of PRC2 via the EED subunit[38], and PRC2 simultaneously contacts both substrate and activating nucleosomes[39], disruption of polycomb spreading by low frequency incorporation of mutant nucleosomes could potentially contribute to this effect, at least for the H3R26 mutant. Why the loss of the PRC2 product is restricted to specific genomic regions remains to be understood.

Prior to this work, the only cancer-associated histone mutation known to significantly reduce H3K27 methylation was the *trans*-acting H3.3K27M oncohistone. In contrast, different oncohistone mutations

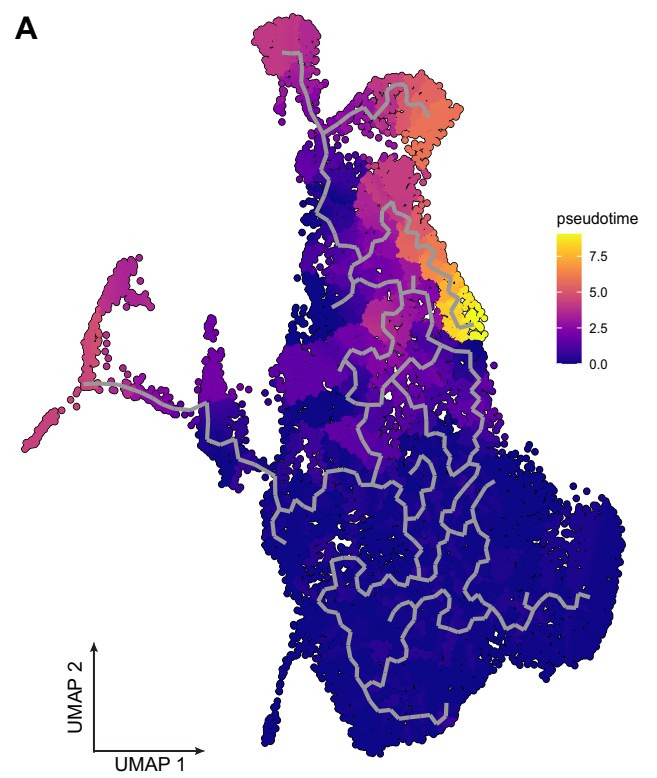

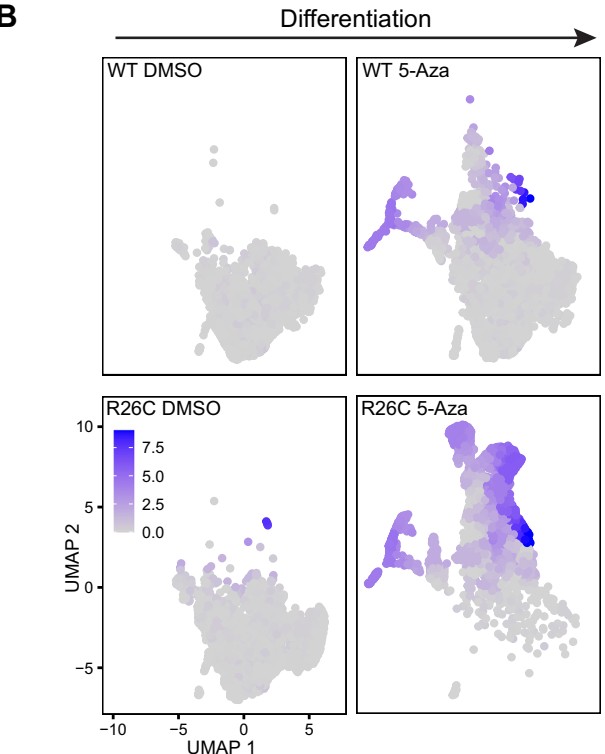

**Fig. 7 | H3R26C expressing MPCs follow an aberrant differentiation trajectory compared to MPCs expressing H3WT. A** Trajectory analysis of the same single cell RNA-seq data from Fig. 6 demonstrating a differentiation path (gray line) and distance in pseudotime (gradient scale shown). **B** Individual samples (defined by genotype and treatment) were projected onto the UMAP with pseudotime indicated by the gradient scale shown.

were recognized to diminish H3K36me3 via either *trans* (H3.3K36M) or *cis* (H3.3G34 mutants) mechanisms. Our findings establish H3R2C and R26C as *cis*-acting oncohistone counterparts of the classical H3.3K27M oncohistone. The effect of the *trans*-acting oncohistones is widespread loss of the affected histone methyl mark, whereas the effect of the *cis*-acting mutants, including H3.3G34 mutants, and based on our observations, H3R2C and R26C, is restricted to a smaller set of histone methylation domains[16,40]. Notably, in all cases, *cis*-acting oncohistone mutants induce locus-specific reorganization of chromatin.

That *cis*- and *trans*-acting oncohistone mutations affecting the same histone PTM are observed in different tumor types[18] raises the possibility that the oncogenic advantages of the resulting magnitude of chromatin perturbation are context-dependent. A strong switch-like effect of a *trans*-acting oncohistone or the more restricted dimmer-like like effect of a *cis*-acting oncohistone could be positively selected depending on the cell of origin, developmental stage, epigenetic state, or background of co-occurring genetic events.

A possible mechanism by which a *cis*-acting PRC2 inhibitors like H3R2C or R26C could potentiate malignant transformation is by establishing a cell population that is susceptible to subsequent oncogenic events. For example, through reduction of H3K27me3, H3R2C and R26C stimulated de-repression of programs associated with lineage commitment in a mesenchymal progenitor cell context. Phenotypically, differentiation of mesenchymal progenitors expressing H3R26C resulted in cell populations that expressed both progenitor and early lineage commitment programs. This mutant also impaired lineage specification in teratomas derived from mESCs, which is in keeping with the effects of PRC2 loss of function in impairing mature differentiation of teratomas[41,42] thereby supporting the proposed mechanism of impaired PRC2 activity by H3R26C. It is possible that the aberrant populations that arise upon differentiation of the H3R26C mutant could potentially serve as primed substrates for further transformation by the action of subsequent driver genetic events, a possibility that warrants further exploration.

While this work focused on the similar effects of the H3R2 and R26 mutants on PRC2 complex activity, there are also important differences between them. Notably, H3R2C reduced H3K4me3 *in cis* implying an effect on the writing and/or erasing of this mark though with the caveat that interpretation of the immunoblot results could be confounded by epitope occlusion. How this factors into the downstream functional effect of this mutant in the tumors expressing it remains to be investigated. We also observed interesting effects in the H3R8 mutant, which have yet to be fully explored. For example, expression of this mutant correlated with marked reduction in the *in cis* prevalence of H3K4me3 and H3K9me3 raising the possibility that it can reduce both active and repressive histone PTMs. Whether this leads to significantly reorganized chromatin states, is one of several open questions.

## Methods

This research was performed in accordance with all relevant ethical regulations; Mice were treated in accordance with a protocol approved by the Rockefeller University Institutional Animal Care and Use Committee.

### Plasmids and lentivirus production

The human H3C2 cDNA sequence was cloned into the pCDH-EF1-MCS-puro lentiviral vector with C-terminal HA- and FLAG-epitope tags[13]. This construct was then used as a substrate for PCR-based site-directed mutagenesis using the Q5 site-directed mutagenesis kit (New England Biolabs) following the manufacturer's protocol to generate histone point mutants H3.1 R2C, R2A, R2H, R8C, R8A, R8H, R17C, R17A, R17H, R26C, R26A, R26A, R26H. The presence of the correct mutations was confirmed by Sanger sequencing. To generate lentiviral particles, HEK293T cells were transfected with the pCDH-EF1-MCS-puro lentiviral

vector harboring the H3.1 insert of interest along with helper plasmids (psPAX2 and pVSVG). The virus containing supernatant was collected in standard culture media containing 1% BSA on day two and filtered.

### Cell lines, cell culture, and generation of stable cell lines

HEK293T and C3H10T1/2 (Clone 8, CCL-26, ATCC) cells were cultured in Dulbecco's modified Eagle medium (DMEM; Gibco) with 10% fetal bovine serum (CellGro) and 1% penicillin/streptomycin (Gibco). V6.5 mouse ES cells (C57BL/6 × 129S4/SvJae $F_1$)[43] (mESCs) were grown on gelatin-coated tissue cultures dishes in Knockout DMEM (Gibco) with 15% ES qualified FBS (Sigma), 0.072% beta-mercaptoethanol, 2mM L-glutamine (Gibco), MEM non-essential amino acids (from 100X stock; Gibco), and LIF (GeminiBio). Cell lines were tested for mycoplasma contamination. To generate cell lines expressing transgenic H3.1, cells were transduced with lentivirus with 5 μg/mL polybrene (Millipore). After 48 h, transduced cells were grown under selection with puromycin (2 μg/mL).

### Arginine methyltransferase inhibitor treatment

Cells were seeded at $1 \times 10^6$ cells per 15 cm dish and allowed to recover overnight. The following day, the cells were treated with 300 nM MS023 (Selleck, S81112), 300 nM GSK591 (Selleck, S8111), 300 nM MS023 and 300 nM GSK591, or vehicle (DMSO). The final DMSO concentration was 0.5% for all conditions. After 2 days of treatment, the cells were collected, washed in PBS, and flash frozen for further analysis.

### Immunoblotting

Protein samples were separated by SDS-PAGE and transferred to a PVDF membrane (or to nitrocellulose for blots of whole cell extracts with anti-methylarginine or beta-actin antibodies and and for anti-H3K4me3 from histone extracts), which was subsequently blocked with 5% milk solution in tris-buffered saline with 0.1% Tween-20 (TBST). Membranes were then probed with primary antibodies in 1% milk (or 5% BSA for anti-methylarginine and beta-actin blots, and for H3K27me3 blots from histone extract) in TBST overnight at 4 degrees Celsius, washed 3X with TBST and incubated with horseradish peroxidase-conjugated secondary antibody for detection using a chemiluminescent substrate (ECL, Pierce). Primary antibodies used were: anti-H3 (Abcam, ab1791; 1:3000-25000), anti-H3K4me3 (Active Motif, 39159; 1:1000), anti-H3K9me3 (Abcam, ab8898; 1:1000), anti-H3K27me3 (Cell Signaling, 9733; 1:1000-3500), anti-H3K27ac (Active Motif, 39133; 1:1000), anti-H3K27me1 (Active Motif, 61015; 1:1000), anti-H3K27me2 (Cell Signaling, 9728; 1:1000), anti-H3K4me1 (Abcam, ab8895; 1:1000), anti-H3K4me2 (abAbcam, 7766; 1:1000), anti-HA (Biolegend, 901503; 1:1000-5000), anti-SDMA (Cell Signaling, 13222; 1:1000), anti-ADMA (Cell Signaling, 13522; 1:1000), anti-MMA (Cell Signaling, 8015; 1:1000), and anti-beta-actin, (Cell Signaling, 4970; 1:1000). Secondary antibodies: anti-mouse IgG, HRP-linked (Cyvita, NA931; 1:5000), anti-rabbit IgG, HRP-linked (Dako, PO399; 1:5000), anti-rabbit IgG, HRP-linked (Cell Signaling, 7074; 1:2000-3000).

### Histone acid extraction

Flash frozen pellets of $1$-$2 \times 10^6$ PBS-washed cells were thawed on ice and incubated on a rotator for 30 min at 4 °C in 1 mL of hypotonic lysis buffer (10 mM Tris-HCl, pH 8.0, 1 mM KCl, 1.5 mM MgCl$_2$, 1 mM DTT, and 1x protease inhibitor cocktail (Roche)). Nuclei were pelted by centrifugation at $10,000 \times g$ at 4 °C, resuspended in 0.4 N H$_2$SO$_4$ then incubated on a rotator at 4 °C for 30 min. The samples were then centrifuged at $16,000 \times g$ at 4 °C for 10 min. The supernatant was then incubated with 132 μL trichloroacetic acid (100%), which was added in a dropwise fashion, for 30 min at 4 °C. The samples were then centrifuged at $16,000 \times g$ at 4 °C for 10 min and the supernatant was discarded. The pellet was washed twice with 1 mL of ice-cold acetone, air-dried, and then resuspended in ddH$_2$0.

## Nucleosome immunoprecipitation

Cell pellets of $3 \times 10^7$ HEK293T cells expressing transgenic epitope tagged histone H3.1 were collected and resuspended in 1 mL of LB1 (50 mM HEPES pH 7.5, 140 mM NaCl, 1 mM EDTA, 10% glycerol, 0.5% NP-40, 0.25% Triton-X-100) with 1× protease inhibitor cocktail (Roche). After a 10-min incubation on a rotator at 4 °C, tubes were spun at $1350 \times g$ at 4 °C and the supernatant was discarded. The pellet was resuspended in 1 mL of LB2 (10 mM Tris-HCl pH 8.0, 200 mM NaCl, 1 mM EDTA, 0.5 mM EGTA) with 1× protease inhibitors and incubated on a rotor at 4 °C for 10 min. After a 5-min spin at $1350 \times g$ at 4 °C, the supernatant was discarded and the pellet was resuspended in 900 μL of LB3 (10 mM Tris-HCl, pH 8.0, 100 mM NaCl, 1 mM EDTA, 0.5 mM EGTA, 0.1% sodium deoxycholate, 0.5% N-laurolysarcosine) with 1× protease inhibitors and 100 μL of 10% Triton-X-100. In series, the sample was passed through 21 G, 23 G, and 27 G needles on ice to produce a homogenous mixture, which was then sonicated using a Covaris E220 ultrasonicator at a peak power of 220, duty factor of 5.0, and cycle/burst ratio of 200. After a 10-min spin at maximum speed in a 4 °C benchtop microcentrifuge, the supernatant was collected and transferred to tubes containing FLAG-conjugated magnetic beads (Pierce A36797) that had been equilibrated with PBS with 1× protease inhibitor cocktail. Following an overnight incubation at 4 °C on a rotator, the supernatant was removed and the beads were washed three times with 900 μL LB3 with 1× protease inhibitor cocktail and 100 μL of 10% Triton-X-100. Three additional washes with Buffer D (20 mM HEPES pH 7.9, 10% glycerol, 0.2 mM EDTA, 0.2% Triton-X-100 and 1× protease inhibitor cocktail) with 100 mM NaCl and then three washes with PBS with 1× protease inhibitor cocktail were performed prior to resuspending the beads in 100 μL of 100 mM glycine pH 2.0 and shaking at 1400 rpm at room temperature to elute the bound proteins (repeated for two elutions total). The eluate was then neutralized by vortexing with 15 μL of 1 M Tris-HCl pH 8.5.

## Middle down histone sample preparation

Elutions from the FLAG- immunoprecipitation were dried down in a speed-vac, and then resuspended in 5 mM dithiothreitol (Thermo Fisher Scientific) in 50 mM ammonium bicarbonate buffer pH 8 and reduced for 1 h at room temperature. Iodoacetamide (Sigma) was added to a final concentration of 20 mM, and samples were alkylated for 30 min at room temperature in the dark. Samples were again dried down in the speed-vac, and then digestion was performed as previously described[44,45]. Briefly, samples were resuspended in 5 mM ammonium acetate pH 4 and digested with endoproteinase GluC at a ratio of 1:20 (enzyme:protein) overnight at room temperature. Digested samples were dried down and then desalted using in-house stage tips[44,45]. Stage tips were generated by wedging a 0.5 cm circular punch of a 3 M Empore C18 paper disk into the bottom of a P200 pipette tip, followed by addition of porous graphitic carbon (PGC) (HyperCarb, Thermo Fisher Scientific) to a volume three times that of the C18 punch. The stage tips were conditioned with acetonitrile (ACN), equilibrated with 0.1% trifluoroacetic acid (TFA), followed by sample loading in 0.1% TFA, then washed with 0.1% TFA, and eluted with 0.1% TFA in 70% ACN. Two biologic replicates were analyzed for each of the H3WT control and H3R26C samples.

## Middle down histone mass spectrometry analysis

The histone samples were analyzed by nanoLC-MS/MS with a Dionex-nanoLC coupled to an Orbitrap Fusion mass spectrometer (Thermo Fisher Scientific). The column was packed in-house using a 75 μm ID × 17 cm column with PGC HyperCarb (3 μm; Thermo Fisher Scientific). The HPLC buffers were A = 0.1% formic acid; B = 80% acetonitrile, 0.1% formic acid. The HPLC gradient was as follows: 2% solvent B for 25 min, then 2% to 13% solvent B in 2 min, from 13% to 16% solvent B in 45 min, 16% to 23% solvent B in 45 min, up to 95% B in 1 min, 95% B for 3 min, and back down from 95% B to 2% B in 1 min and a hold at 2% B for

9 min. The flow rate was at 500 nL/min. Data were acquired using a data-dependent acquisition method, consisting of a full scan MS spectrum (m/z 350–900) performed in the Orbitrap at 120,000 resolution with an AGC target value of 5e5 (normalized AGC target = 125%), followed by selection of peptides of charge states 7–12 and MS2 fragmentation using ETD with calibrated charge-dependent parameters. Dynamic exclusion was set to 10 seconds. Fragmented peptides were detected in the Orbitrap at 30,000 resolution. AGC target was set at 7.5e4 (normalized AGC target = 150%), and maximum inject time set at 200 ms. Histone samples were resuspended in 10 μl buffer A and 4 μl was injected.

## Middle down mass spectrometry data analysis

Middle down MS data analysis was performed as previously described[44,45]. Briefly, raw files were processed in Proteome Discoverer 2.2 (Thermo Fisher Scientific) using the Mascot search engine with a precursor mass tolerance of 2.1 Da and a fragment mass tolerance of 0.01 Da. Carbamidomethyl on cysteine was set as a fixed modification, and variable modifications included acetylation on n-termini and lysine, mono- and di- methylation on lysine and arginine, and trimethylation on lysine. Mascot output files were analyzed with in-house software, HistoneCoderTool ProteoformQuant and IsoScale[46,47] (http://middle-down.github.io/Software/), which removes peptides that do not have sufficient fragment ions to map PTMs, and only retains unambiguous peptides with high confidence identification and quantification of PTMs. The output file contains the quantified relative abundance for every identified peptide species, as well as the MS2 level evidence for PTM identification and localization.

## Synthesis of Fmoc-Lys(me3)-OH-TFA

Fmoc-Lys(Boc)-OH (6 g) was deprotected by addition of 1:1 trifluoroacetic acid:dichloromethane with stirring for 1 h at room temperature. The solution was concentrated in vacuo overnight and the resultant residue resuspended in 4:1 dichloromethane:MeOH followed by slow addition of N,N-diisopropylethylamine (15 eq.) and then methyl iodide (15 eq.). Reaction progress was monitored by RP-HPLC and quenched by addition of water after 70 min. The product was subjected to a series of extractions with dichloromethane and the final aqueous layer acidified with trifluoroacetic acid to pH 2 and purified by preparatory HPLC. The final product was obtained in 51% yield and characterized by RP-HPLC, ESI-MS, and 1H NMR spectroscopy.

## Synthesis of wild-type and arginine mutant H3K27me2 and H3K27me3 histones

Wild-type H2A, H2B, and H4 histones and truncated histone H3.1$_{29-135}$(A29C, C96A, C110A) were recombinantly expressed and purified from E. coli. Peptide hydrazides for semisynthesis were prepared by solid-phase peptide synthesis as previously described[48] with Fmoc-Lys(alloc)-OH incorporated at the desired locations for Lys(me2). For synthesis of H3K27me2-containing constructs, peptides were deprotected on resin with Pd(PPh$_3$)$_4$ (0.25 eq.) and N,N-dimethylbarbituric acid (10 eq.) in dichloromethane with agitation (2 × 30 min) and then washed with N,N-diethyldithiocarbamate in dimethylformamide. Resin was subsequently washed with methanol (MeOH) and equilibrated in a 1:1 mixture of PBS:MeOH. Reductive alkylation was performed by addition of 37% (w/w) formaldehyde (50 eq.) in 1:1 PBS:MeOH and solid NaCNBH3 (50 eq.) with agitation (2 × 30 min). Samples were then washed with 1:1 PBS:MeOH, MeOH, and dimethylformamide (sequentially) and peptide cleavage performed as previously described[48]. Peptide hydrazides were dissolved in activation buffer (6 M guanidinium hydrochloride, 0.2 M NaH$_2$PO$_4$, pH 3.0) and reacted with a solution of NaNO$_2$ (10 eq.) in a salt/ice slurry at −20 °C for 20 min. 2-Mercaptoethanesulfonic acid sodium salt (100 eq.) was then added and the solution pH adjusted slowly to pH 7, after which the reaction was allowed to proceed at room temperature for 15–20 min. The

resultant thioesters were purified by semipreparatory RP-HPLC as previously described[48]. Peptide sequences for histone semisynthesis are shown below:

H3(1-28)K27me2: ARTKQTARKSTGGKAPRKQLATKAARK(me2)S
H3R2A(1-28)K27me2: AATKQTARKSTGGKAPRKQLATKAARK(me2)S
H3R8A(1-28)K27me2: ARTKQTAAKSTGGKAPRKQLATKAARK(me2)S
H3R17(1-28)K27me2: ARTKQTARKSTGGKAPAKQLATKAARK(me2)S
H3R26A(1-28)K27me2:        ARTKQTARKSTGGKAPRKQLATKAAAK
(me2)S

H3(1-28)K27me3: ARTKQTARKSTGGKAPRKQLATKAARK(me3)S
H3R26A(1-28)K27me3:        ARTKQTARKSTGGKAPRKQLATKAAAK
(me3)S

For native chemical ligation, co-lyophilized purified peptide thioesters (2 mg) and truncated histone H3.1$_{29-135}$ (2 mg) were resuspended in ligation buffer (6 M guanidinium hydrochloride, 100 mM Na2HPO$_4$, 10 mM TCEP, 10 mM N-acetyl methionine, pH 7.5) with 5% (v/v) 2,2,2-trifluoroethanethiol (TFET). The pH of the solution was carefully adjusted to pH 7.5 and incubated at 37 °C for 4 h, with reaction progress monitored by RP-HPLC. Upon completion, the ligation reaction was diluted two-fold by addition of desulfurization buffer (400 mM TCEP and 40 mM reduced glutathione), degassed, and carefully pH adjusted to pH 7.0. Radical initiator VA-044 was added from a 20x aqueous stock to give a final solution concentration of 10 mM and the reaction allowed to proceed overnight at 37 °C. Reaction progress was monitored and products purified by semipreparatory RP-HPLC. All final histones were characterized by C18 RP-HPLC and ESI-MS.

## Preparation of nucleosome arrays

12 x 601 DNA (DNA containing 12 repeats of the 147-bp 601 sequence with 30-bp linkers) was purified from DH5α *E. coli* as previously described[49] with an additional prep cell electrophoresis purification step (Model 491 Prep Cell, Bio-Rad).

601 DNA sequence: ACAGGATGTATATATCTGACACGTGCCTGG
AGACTAGGGAGTAATCCCCTTGGCGGTTAAAACGCGGGGGACAGCGC
GTACGTGCGTTTAAGCGGTGCTAGAGCTGTCTACGACCAATTGAGCG
GCCTCGGCACCGGGATTCTCCAG

MMTV DNA was prepared by an analogous method, with precipitation by addition of 7.5% (v/v) PEG-6000 used for purification (resultant 155-bp MMTV DNA fragment remains in the supernatant).

MMTV DNA sequence:

ACTTGCAACAGTCCTAACATTCACCTCTTGTGTGTTTGTGTCTGT
TCGCCATCCCGTCTCCGCTCGTCACTTATCCTTCACTTTCCAGAGGG
TCCCCCCGCAGACCCCGGCGACCCTGGTCGGCCGACTGCGGC ACAG
TTTTTTG

Histone octamers were assembled and FPLC-purified as previously described[21]. 12-mer nucleosome arrays were assembled as previously detailed[48] with minor modifications, namely that arrays were precipitated with 4 mM MgCl$_2$ and resuspended array pellets were dialyzed against array assembly end buffer (200 mL × 1 h at 4 °C) to remove MgCl$_2$ from the final array preparations. Arrays were quantified by UV spectroscopy at 260 nm and assembly assessed by agarose polyacrylamide gel electrophoresis with ethidium bromide staining.

## PRC2 activity assay

PRC2 core complex (PRC2), consisting of subunits EZH2, EED, RBBP4, and SUZ12, was expressed and purified from Sf9 cells as previously described[48]. PRC2 methyltransferase activity was assessed by radiometric assay in which 12-mer nucleosome arrays (480 nM 601 sites) were incubated with 20 nM PRC2 in 10 µL histone methyltransferase assay buffer (50 mM HEPES, pH 8.0, 35 mM NaCl, 0.5 mM MgCl$_2$, 0.1% (v/v) Tween-20, 5 mM DTT, 1 mM PMSF, and 0.66 µM [3H]-S-adenosylmethionine) for 1 h at 30 °C. Reactions were quenched by spotting onto Whatman P81 phosphocellulose filter paper (Sigma). Filter papers were air dried for 45 min and then washed 3 × 15 min with 0.2 M NaHCO$_3$ (pH 9.0) with shaking. Filter papers were subsequently dried at 40 °C for 1 h using a gel dryer, submerged in 1 mL Ultima Gold scintillation cocktail, and incubated overnight with shaking. Scintillation counting was performed the following day on a MicroBeta scintillation counter (PerkinElmer), with experimental sample counts corrected for background using a reaction in the absence of enzyme.

## CUT&RUN assay

C3H10T1/2 cells were collected by incubation with TrypLE reagent (Gibco) for 2 min at 37 °C followed by quenching with culture media and centrifugation. Cell pellets were resuspended in 1 mL PBS and transferred to a 2 mL screw cap tube. The CUT&RUN protocol was performed in PCR-tubes per the CUTANA (Epicypher) v1.6 protocol except where otherwise noted. In brief, cells were washed twice in wash buffer (20 mM HEPES, 150 mM NaCl, 0.5 mM Spermidine, EDTA-free Protease Inhibitor) prior to resuspending wash buffer. For each CUT&RUN reaction, 500,000 cells were immobilized to Concanavalin-A beads and incubated overnight at 4 °C with antibody diluted 1:100 in antibody dilution buffer (0.005% Digitonin, 2 mM EDTA, Wash Buffer). pAG-MNase (Epicypher, 15–116) digestion was performed for 2 h at 4 °C following which CUT&RUN enriched DNA was purified using phenol-chloroform extraction and ethanol precipitation. NEBNext Ultra II DNA Library Prep kit (NEB, E7645S) was used to prepare sequencing libraries with CUT&RUN enriched DNA. Libraries were sequenced using the Illumina NextSeq 550. Antibodies used were H3 (Abcam, 1791, HA (Biolegend, 901501), H3K4me3 (Active Motif 39159), H3K27me3 (Cell Signaling, 9733), and rabbit anti-mouse IgG (Abcam, 46450).

## CUT&RUN analysis

Sequence and transcript coordinates for mouse mm10 and gene models were retrieved from the Bioconductor Bsgenome.Mmusculus.UCSC.mm10 (version 1.4.0) and TxDb.Mmusculus.UCSC.mm10.knownGene (version 3.4.0) Bioconductor libraries respectively.

For the analysis of CUT&RUN data, reads were mapped using the Rsubread package's align function (version 1.30.6)[50]. Peak calls made with SEACR (version 1.3, stringent, norm 0.01)[51]. Consensus peaks were determined to be peaks that were found in the majority of replicates in at least one mutant or control. Peaks were annotated and genome distribution was determined using the ChIPseeker package (Version 1.28.3)[52]. Enrichment analysis was performed using clusterProfiler (Version 4.0.2)[53] against all gene sets from msigdbr (Version 7.4.1) that were related to polycomb or H3K27me3. Peaks were annotated to genes based on proximity; peaks that overlapped with TSS regions were considered for comparisons. Pairwise comparisons were made between the control and mutant using DEseq2 using counts from consensus peaks with significant genes considered as (padj <0.001). Normalized, fragment-extended signal bigWigs were created using the rtracklayer package (Version 1.40.6), and then visualized and exported from IGV.

Ranged heatmaps were generated with profileplyr (Version 1.12.0). Volcano plots were drawn with EnhancedVolcano (Version 1.10.0).

Motif analysis was completed using the MEME suite (Version 5.4.1)[54] and its wrapper for R, memes (Version 1.04). Motif enrichment was tested using AME from MEME within the promoter regions of genes that lose H3K27me3 (−1/+0.2 Kb). All promoter regions were used as a background for this test. The CIS-BP database was the source of known motifs (version 2.0).

## Bulk RNA-seq and analysis

C3H10T1/2 cells from three separate cultures on different days were collected by washing with PBS, incubation with TypLE for 10 min at

37 °C followed by quenching with culture media and centrifugation at 300 × *g*. The pellet was resuspended in ice cold PBS, collected at 300 × *g* at 4 °C, followed by aspiration and flash freezing in liquid nitrogen followed by storage at −80 °C. RNA was then extracted using the RNAeasy kit (Qiagen) with on column DNAase I digestion per manufacturer instructions. From each sample, 500 ng of RNA was used to generate Illumina sequencing libraries using the NEBNext Ultra II RNA Library Preparation Kit (New England Biolabs) with polyA selection according to the manufacturer instructions. Libraries were pooled and single end sequencing (75 bp) was performed using an Illumina NextSeq 500.

Transcript abundance was determined using Salmon (v0.8.1) and the GENCODE reference transcript sequences[55]. Transcript counts from Salmon were imported into R with the tximport R Bioconductor package (v1.20), and differentially expressed genes were determined with the DESeq2 R Bioconductor package (v1.32)[56,57]. Read counts were normalized using the rlog function from DESeq2 and z-scores for the indicated gene sets were visualized using the ComplexHeatmap R Bioconductor package (v2.8)[58]. For GSEA analysis, genes were ranked using the Wald statistic as calculated by DESeq2 and then compared against the indicated gene lists using the clusterProfiler R Bioconductor package (v4.0.5)[59,60]. Hallmark gene lists were obtained from the Molecular Signatures Database using the msigdbr R package (v7.5.1) and GSEA results were visualized using the enrichPlot Bioconductor R package (v1.12.3)[60,61]. Gene Ontology analysis was performed using TopGO (v2.40.0) and visualized with Python (see data availability section for access to code).

For the analysis of RNAseq data used in Fig. 4C, transcript expressions were calculated using the Salmon quantification software (version 0.8.2)[55]. Normalization and differential gene expression analysis were performed using DESeq2 with significant genes considered as ($p_{adj} < 0.05$) (version 1.20.0)[62].

### Mesenchymal differentiation assay

C3H10T1/2 cells expressing H3WT or H3R26C were cultured in Dulbecco's Modified Eagle Medium (Corning) with 10% fetal bovine serum, 2 µg/ml puromycin, and 1% penicillin-streptomycin. For the 5-Azacytidine differentiation assay, cells were seeded in 2-well chamber slides (Ibidi, 80286) with 190 cells per chamber. Cells were treated with 3 µM 5-Azacytidine (Sigma A2385) for 24 h following which media was replaced twice a week. After 4 weeks, cells were processed for immunofluorescence. The results from the 5-Azacytidine differentiation assay were analyzed across 4 biological replicates.

### Differentiation assay immunofluorescence staining and image capture

Cells undergoing differentiation were fixed with 4% Formaldehyde, washed in PBS and permeabilized (0.1% Triton X-100 for Myosin 4, 0.2% Digitonin for LipidTOX staining). Following permeabilization, cells were blocked with 3% BSA in PBS and incubated overnight at 4 °C with Myosin 4 antibody (Invitrogen14650382, 1 µg/ml) in antibody dilution buffer (2% BSA in PBS), or alternatively incubated for 1 h at RT with LipidTOX (Invitrogen, H34475, 1:200). After overnight incubation with the Myosin 4 antibody, cells were washed in PBS and incubated for 1 h at RT with anti-mouse Alexa Fluor 568 secondary antibody (Invitrogen, A-11004, 2 µg/ml). Cells were washed in PBS and stained with DAPI (Sigma D9542, 2 ug/ml) for 5 min. Samples were rinsed with PBS and Vectashield antifade mounting media (Vector Laboratories, H-1000) was added to the chamber slides. Fluorescence images were taken by imaging the entire slide using Zeiss Celldiscoverer 7 (CD7) automated widefield system with DAPI (Excitation/Emission 353/465 nm), AF488 (Excitation/Emission 493/517 nm), and AF568 (Excitation/Emission 577/603 nm) channels. ZEN 2.6 (Blue edition) software was used for image acquisition.

### Differentiation assay immunofluorescence image analysis

ZEN 2.6 (blue edition) was used to stitch tiles and create maximum intensity projections from the original.czi Z-stacks. For quantifying adipogenic differentiation, images with LipidTOX staining were analyzed. For computational efficiency, the entire image was divided into four non-overlapping sub-images. For each sub-image (≈12,000 × 10,002 microns), the area covered by LipidTOX droplets was calculated using the Bernsen Method of Auto Local Threshold (Param1 = 55, Param2 = 0) command in Fiji (version 2.3.0/1.53). The total number of nuclei in each sub-image was calculated using a custom StarDist script in QuPath (version 0.3.0). The total area covered by LipidTOX droplets in the entire image was calculated as the sum of areas covered by LipidTOX in each sub-image. Similarly, the total number of nuclei in the entire image was calculated as the sum of the number of nuclei in each sub image. Adipogenesis quantification was adapted from established methods[35] and was quantified as the Adipogenic Index (AI), which is expressed as the percentage area covered by LipidTOX normalized to the total number of nuclei for each sample (arbitrary units) (Supplementary Fig. S8). Average values of AIs across 4 biological replicates were plotted and ratio paired t-test was performed for statistical analysis.

For quantifying myogenic differentiation, four non-overlapping regions (≈7955 × 4290 microns) with Myosin 4 signal were analyzed for each sample. These regions were analyzed in QuPath software (version 0.3.0). Myosin 4 positive cells were outlined manually by using the annotation tool in QuPath and number of nuclei in each Myosin 4 positive cell annotation was quantified using a custom StarDist script. The total number of Myosin 4 positive cells and the number of Myosin 4 positive cells with more than one nuclei was used to calculate the percentage of multinucleated Myosin 4 positive cells. Average values of the percentage of multinucleated Myosin 4 positive cells across 4 biological replicates were plotted and a Mann–Whitney test was performed for statistical analysis.

### Histone mutant immunofluorescence and imaging analyses

HEK293T cells stably expressing HA-tagged wild type H3.1 and a panel of arginine mutants were grown in 12-well plates with 18 mm coverslip placed on the bottom (Neuvitro GG-18-1.5). Upon reaching 50–75% confluence, cells were rinsed in PBS, fixed with 1% paraformaldehyde in 0.1% Triton X-100 in PBS (PBS-T) for 20 min, washed with PBS-T and blocked for one hour in 1% BSA in PBS-T. Coverslips were then incubated with primary antibodies to HA epitope (mouse HA.11, BioLegend 901503, 1:200) and total H3 (rabbit ab1791, abcam, 1:1000) in 1% BSA-PBS-T solution in humidity chambers overnight, washed three times with PBS-T, and incubated with AlexaFluor-conjugated secondary antibodies (Invitrogen donkey anti-rabbit AF488 (A21206) and goat anti-mouse AF568 (A11031)) diluted 1:1000 in 1% BSA-PBS-T for four hours. Coverslips were then washed twice with PBS-T, incubated for 10 min with DAPI (1 µg/ml final concentration) in PBS, rinsed with PBS, and mounted on glass slides with ProLong Gold reagent (Invitrogen P36934). Slides were imaged using Zeiss LSM 780 AxioObserver, C-Apochromat 40x/1.2 water objective, and Zen suite (Rockefeller University BioImaging Resource Center). Single confocal slices were minimally processed in ImageJ and assembled in Adobe Illustrator CC.

### Single cell RNA-seq and analysis

C3H10T1/2 cells expressing H3WT or H3R26C were differentiated using 5-Azacytidine as described above. On day 28, cells were washed with PBS and dissociated using TrypLE Express, which was quenched with Dulbecco's Modified Eagle Medium supplemented with 10% fetal bovine serum. The cells were collected by centrifugation and resuspended in 0.04% BSA in PBS. The cells were passed through a 70 µm cell strainer and 50,000 cells were submitted to the Memorial Sloan Kettering Integrative Genomics Operation core facility for further processing and sequencing on a 10X genomics platform.

For analysis, the FASTQ files were first processed using CellRanger V6 with the mouse genome and transcript reference data (refdata-gex-mm10-2020-A) supplied by 10X genomics. The output from CellRanger was then analyzed using the Seurat R package version 4. The filtered barcode/feature matrix was read using Seurat read10X command and the percentage of mitochondrial reads per cell was computed for subsequence filtering. Cells were filtered using the following criteria: percent mitochondrial read greater than 20% or the number of genes was less than 1500 or the number of distinct molecules was less than 5000. If any of these conditions were true the cell was filtered out of the downstream analysis. Next, we scored the cell cycle phase of each cell using Seurat's scoreCellCycle functions. The list of cell cycle genes for mouse was determined by taking the supplied list of human cell cycle genes and mapping them to mouse genes using the bioMart R package to do the homology mapping.

The filtered data was normalized and scaled using the SCTransform method from Seurat. For the transform we regressed against the cell cycle scores previously computed. After normalization we computed the PCA coordinates and retained the first 20 coordinates in the clustering and projection analysis. For clustering we used the Seurat FindNeighbor and FindClusters functions with several resolution values and after manual inspection fixed on a resolution value of 0.2 for subsequent work. We also computed the UMAP project using RunUMAP and 20 pca coordinates. Cluster specific marker genes were computed using FindAllMarkers with a cutoff of 0.25 in the log fold change and a minimum percentage of 25%. For two specified gene sets: Adipocyte[63] and Skeletal Muscle [https://www.gsea-msigdb.org/gsea/msigdb/cards/SKELETAL_MUSCLE_DEVELOPMENT] (Supplementary Data 10), we computed a score for each using AddModuleScore.

Trajectory analysis was done using Monocle3 (version 1.3.1). We used the R package SeuratWrappers to convert the Seurat objects for use in Monocle preserving the original pca mapping and UMAP reduction. The data was re-clustered with Monocle's cluster_cells function and then the trajectory graph and cell pseudotimes were computed.

All of the custom R scripts used in this analysis are available as described in the code availability section. The following software packages were also used: CellRanger Version 6 (https://support.10xgenomics.com), R Version 4.1.2 (https://www.r-project.org), Seurat Version 4.2.0 (https://satijalab.org/seurat), Monocle3 Version 1.3.1 (https://cole-trapnell-lab.github.io/monocle3), SeuratWrappers Version 0.3.1, and SeuratObject Version 4.1.3.

## Teratoma formation and histologic analysis
To generate teratomas, $1 \times 10^6$ mESC expressing H3WT or H3R26C were injected subcutaneously in each flank of 6–7 weeks old female NOD.Cg-Prkdcscid immunodeficient mice (Jackson Laboratories, Strain 005557). The mice were housed in ambient temperatures from 68–79 degrees Fahrenheit, 30–70% humidity, with light/dark times of 0700/1900 hours. For subcutaneous injections, mESC were mixed 1:1 with Matrigel Basement Membrane Matrix (Corning, 356231) and a total volume of 200 µL was injected in each flank. After 3 weeks, tumors were excised immediately postmortem and fixed in 10% Neutral Buffered Formalin (v/v). Mice were treated in accordance with a protocol approved by the Rockefeller University Institutional Animal Care and Use Committee. The maximal allowed tumor size/burden was 1.5–2 cm or ulceration; this limit was not exceeded. Fixed tumors were submitted to the Laboratory for Comparative Pathology at the Memorial Sloan Kettering Cancer Center for processing, embedding, sectioning, and hematoxylin and eosin staining. Germ layer proportions were assigned by a board-certified veterinary pathologist (SCS). Sex was not considered in the study design or analysis since the experiments were designed and controlled to test the effect of a histone mutation as the variable of focus.

## Statistical & reproducibility
The statistical methods for analysis are described in the relevant contexts in the methods, figure legends, or embedded in published or custom code (see code availability section). Investigators performing histologic analysis of teratomas were blinded to the genotypes of the mESCs used to generate the teratomas. The experiments were not randomized. Sample size is described in the figure legends and/or methods section. No statistical method was used to predetermine the sample size. No data were excluded from the analyses.

## Reporting summary
Further information on research design is available in the Nature Portfolio Reporting Summary linked to this article.

## Data availability
The sequencing data generated in this study have been deposited in the Gene Expression Omnibus (GEO) database under accession number GSE239638. The mass spectrometry data generated in this study have been deposited in MassIVE database under accession number MSV000092535 [https://massive.ucsd.edu/ProteoSAFe/dataset.jsp?task=5f3e25a5d5f34ac598222b7789ab6f94] (PXD044141). Gene lists were obtained from the Molecular Signatures Database using the msigdbr R package (v7.5.1). The CIS-BP database was the source of known motifs (version 2.0). Other source data are provided with this paper as a Source Data file. Source data are provided with this paper.

## Code availability
Custom code used for analysis is publicly available at https://github.com/soccin/scRNA/tree/proj/p12553 for scRNA-seq analysis and at https://github.com/nacevlab/H3R for all others.

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

## Acknowledgements

This study was supported by NCI K08 CA245212 (BAN) and P01 CA196539 (C.D.A., T.W.M., B.A.G.). B.A.N. is a Damon Runyon Clinical Investigator supported (in part) by the Damon Runyon Cancer Foundation (CI-124-23). We also gratefully acknowledge funding from the NIH (5T32GM071339-13) and the Blavatnik Family Foundation (Y.P.). M.M.M. was supported by an NIH postdoctoral fellowship (GM131632). A.A.S. and C.D.A. were supported by NCI R01 CA234561. B.A.G. was supported by NIH HD106051, NSF grant CHE-2127882 and a St. Jude Children's Research Hospital Chromatin Consortium grant. The Memorial Sloan Kettering Cancer Center Support Grant (P30 CA008748) and Hillman Cancer Center Support Grant (P30 CA047904) partly supported this work through core facility and other resources used in this research. Microscopy was performed in the Rockefeller University's Bio-Imaging Resource Center, RRID:SCR_017791. We thank Ved Sharma, Tao Tong, and Priyam Banerjee from the Rockefeller University's Bio-Imaging Resource Center for help with imaging and image analysis.

## Author contributions

Conception and design: B.A.N., C.D.A. Development of methodology: B.A.N., C.D.A., Y.D., C.P. Data collection: B.A.N., Y.P., M.M.M., Y.D. Y.F., C.P., A.A.S., S.T., M.J.G. Analysis and interpretation of data: B.A.N., M.M.M., A.A.S., C.D.A., Y.D., M.P., T.C., S.M., S.J., D.B., N.S., S.T., M.J.G. Drafting the manuscript: B.A.N. Review and revision of the manuscript: All authors except C.D.A. Visualization of data: B.A.N., A.A.S., Y.D., M.P., T.C., D.B., N.S. Study supervision and/or funding acquisition: B.A.N., C.D.A., T.W.M., B.A.G., W.D.T.

## Competing interests

W.D.T.: Consulting, Advisory Role, Honoraria: Aadi Biosciences, Abbisko, Amgen, AmMAx Bio, Avacta, Ayala Pharmaceuticals, Bayer, BioAlta, Boehringer Ingelheim, C4 Therapeutics, Cogent Biosciences, Curadev, Daiichi Sankyo, Deciphera, Eli Lilly, Epizyme Inc (Nexus Global Group), Foghorn Therapeutics, Ikena Oncology, IMGT, Inhirbix Inc., Ipsen Pharma, Jansen, Kowa Research Inst., Medpacto, Novo Holdings, PER, Servier, Sonata Therapeutics; research funding from Novartis, Eli Lilly, Plexxikon, Daiichi Sankyo, Tracon Pharma, Blueprint Medicines, Immune Design, BioAlta, Deciphera; Patents, Royalties, Other Intellectual Property: Companion Diagnostics for CDK4 inhibitors (14/854,329), Stock and Other Ownership Interests: Certis Oncology Solution, Atropos. All other authors declare no competing interests. There are no patents related to this manuscript.

## Additional information

[1]Department of Medicine, University of Pittsburgh School of Medicine, Pittsburgh, PA 15213, USA. [2]Department of Pathology, University of Pittsburgh School of Medicine, Pittsburgh, PA 15213, USA. [3]UPMC Hillman Cancer Center, Pittsburgh, PA 15213, USA. [4]Laboratory of Chromatin Biology and Epigenetics, The Rockefeller University, New York, NY 10065, USA. [5]Bioinformatics Resource Center, The Rockefeller University, New York, NY 10065, USA. [6]Department of Chemistry, Princeton University, Princeton, NJ 08544, USA. [7]Department of Biochemistry and Biophysics, Perelman School of Medicine, University of Pennsylvania, Philadelphia, PA 19104, USA. [8]Department of Medicine, Memorial Sloan Kettering Cancer Center, New York, NY 10065, USA. [9]Bioinformatics Core, Memorial Sloan Kettering Cancer Center, New York, NY 10065, USA. [10]Laboratory of Comparative Pathology, Memorial Sloan Kettering Cancer Center, New York, NY 10065, USA. [11]Department of Biochemistry and Molecular Biophysics, Washington University School of Medicine, St. Louis, MO 63110, USA. [12]Present address: Department of Neuroscience, Developmental and Regenerative Biology, The University of Texas at San Antonio, San Antonio, TX 78249, USA. [13]Deceased: C. David Allis. ✉e-mail: ben46@pitt.edu

