## [Peer Review File · Nature Communications]

REVIEWER COMMENTS

Reviewer #1 (Remarks to the Author): Expert in chromatin remodelling and histone modifications in cancer, development, and in vivo models

The discovery of 'oncohistones' ~10 years ago revolutionized our understanding of how changes in the fundamental building blocks of chromatin can alter gene expression programs to drive oncogenic transformations. This paper by Nacev et al reports additional 'oncohistone' mutations in histone H3 at R2, R8, and R26. These mutations impact methylation of lysines important for gene regulation, notably H3K4 and H3K27. These affects apparently occur in cis, in the same nucleosomes that carry the mutant histone, but they impact large domains that are normally repressed by PRC2-mediated H3K27me3. The loss of repression in the H3R2 and R26 mutant bearing cells is associated with defective differentiation, consistent with impaired PRC2 functions.

The discovery of additional recurrent cancer associated missense mutations in histone H3 is interesting and impactful. In general, the experiments are well executed and the findings will be of broad interest.

A few suggestions to make the paper even stronger:

1. The use of the term 'second generation' to describe these H3 mutations could cause confusion, as it could be taken to mean these mutations as subsequent to or evolve from the previously defined mutations. The term also downplays the importance of the current findings, suggesting they might be less important than the previously defined mutations. Why not just say the findings here expand the list of known mutations to additional sites, without referring to them as 'second' anything?
2. Understanding the data in Fig 1 is critical to understanding the rest of the paper. As such, the text describing these experiments needs to be more clear. As written, it seems the blots shown are of cell lysates (and that is what is written in the figure legend), but if that is true, there is no way the authors could know if the changes in lysine methylation were occurring in cis with the R mutations. The references mentioned (8-13) and the methods indicate these are likely actually blots of affinity isolations of the tagged histones. Please add a sentence or two to the text to make this clear.
3. One explanation for the effects of the R2 mutant (and the R26 mutant) on PRC-mediated repression is a change in expression of components of the PRC complex itself. Please indicate whether any change in the RNA or protein levels of the PRC components are altered in the R mutants.
4. How do the effects of the R2 or R26 mutations in differentiation compare to the effects of PRC loss? This could be assessed through direct comparison in these cells, or at least discussed more fully in the Discussion.
5. Both R2 and R26 are subject to methylation themselves, but Rme is not mentioned at all in the paper. Could loss of Rme be related to the effects here? Can the authors use artificial nucleosomes with R2me or R26me as substrates for PRC2? At the very least, Rme and PRMT functions should be discussed in the context of possible cross-talk with Kme and PRC (and MLL) functions.

Reviewer #2 (Remarks to the Author): Expert in functional genomics and epigenomics, chromatin modifications, cancer genomics, and development

In this manuscript, Nacev and colleagues focused on the H3 R2/R26 oncohistones with mutation frequencies similar to those of classical at H3K27, G34, or K36. They demonstrated that H3 N-terminal R2 and R26 mutations impaired PRC2 activity *in vivo* and *in vitro*. They further examined H3R2C and R26C mutation-impacted H3K27me3 domains and downstream transcriptional pathways. By using mesenchymal progenitor cells and murine embryonic stem cell-derived teratomas models, they showed H3R26C mutant could affect lineage commitment of these progenitor cells.

Overall, this study expands the repertoire of oncohistones and presents novel findings on H3R2 linked to a remote site (H3K27). However, before its acceptance for publication, the following points should be addressed:

1. An *in vivo* cancer model should be used for the analysis of H3 R2/R26 oncohistones' contribution to carcinogenesis.
2. While arginine methylation at H3 R2 has been well established, and H3K4me3 is severely impaired in all three cell lines with H3R2C mutation (Figure 2A-C), the changes of H3K4me3 and H3R2me2 profiles and their connection with H3K27me3 and downstream pathways in H3R2C cells should be analyzed.

Reviewer #3 (Remarks to the Author): Expert in single-cell RNAseq, development, chromatin modifications, and computational genomics

Nacev et al. have reported findings that two tumor-associated H3 mutations (H3R2C and H3R26C) reduce H3K27me3 in multiple cell lines and disrupt differentiation as evidenced in ESC-derived teratomas and mesenchymal progenitor cells.

While the manuscript is generally well-composed, there are significant concerns regarding genomic data analysis.

Major:

1. In Figure 3B, C, and Supplementary Figure 3, the inferred decline in H3K27me3 signals isn't convincingly evidenced by the heatmaps, although the aggregated curves provide some clarity. It's advisable for the authors to consider scatter plots (with each dot representing one peak) that illustrate per-peak signals (reads per million) between WT and R2C/R26C. Subsequent statistical tests should be conducted to demonstrate the significance of this decrease.
2. The result in Figure 3A shows the overlap between the K27me3 differential peaks and Polycomb Regulated Gene Sets. The method to assign each peak to a gene is not clear. A more direct way is to compare these peaks with Polycomb-bound peaks and use another irrelevant protein's peaks as the control set.
3. The authors claimed that loss of H3K27me3 "occurred more often within 1 kb of promoters" using Figure 3H which doesn't show sample sizes. It may be worth plotting the H3K27me3 signals over the promoter regions and comparing them with the non-promoter peaks to show the difference. This can be coordinated with the first point, denoting promoters in a different color in the scatter plot.

4. The Gene Ontology analysis is rather superficial and many leading terms are generic ones such as cell adhesion and rRNA processing. What are the driving genes underpinning the interesting GO terms the authors have highlighted? Some examples need to be provided.
5. The motif analysis is useful in understanding the pathways associated with H3K27me3 loss but only Arnt was mentioned. What other TFs have been called? The manuscript should elaborate on the top hits, either within the figure annotations or the main body.
6. The single-cell RNA-seq data seem to conflict with the conclusion that R26C disrupts differentiation because in Figure 6D, the R26C 5-aza cells are more peripheral on the UMAP whereas WT 5-Aza is closer to the DMSO control region. The authors should draw arrows to point out the hypothetical differentiation path and explain why they think R26C disrupts differentiation.
7. The authors used the top 20 PCs for scRNA-seq analysis which might overlook more structures. What's the rationale for choosing that? How does it compare with using 50 PCs?
8. Figure 6C shows strong association between certain clusters and sample identities. Have the authors tried Seurat's batch correction to see if the association is from technical effects?
9. How do the findings from the single-cell data correlate with the earlier-mentioned bulk data? Can the differential genes identified in the bulk dataset be parsed into lineage-specific programs using single-cell data?

Minor:

1. The x-axis of Figure 1a lacks clarity. What does the length of each bar mean?
2. Figure 1a and 1b share a y-axis; aligning them at the same height could enhance visual coherence. Alternatively, merging them might be effective—wherein the x-axis of one panel is transposed to the y-axis and dot size indicates mutation count.
3. Figure 2F isn't showing much information when the gene clusters are not annotated. It might be moved to Figure 4?
4. Figure 2E can label the sample IDs for individual data points
5. Line 228's "H2K27me3" should be "H3K27me3"
6. Supplementary Table 6's second tab lacks protein names.
7. For the single-cell analyses in Figure 6, it would be useful to include more intuitive names than cluster IDs, especially in Figure 6C. For example, Group 9 can be called CD34+ MPC etc.

Reviewer #4 (Remarks to the Author): Expert in cancer epigenetics, CUT&RUN, and mass spectrometry

The CUT&RUN assays and data analysis were performed by the standards in the field.

There are only two minor points that the authors should include:

- the number of unique reads per C&R library
- why did the authors choose for a 2 hrs/4°C-pA/G-MNase digestion rather than for 30 min/0°C.

Response to reviewers

Manuscript ID:

Reviewer #1 (Remarks to the Author): Expert in chromatin remodelling and histone modifications in cancer, development, and in vivo models

The discovery of 'oncohistones' ~10 years ago revolutionized our understanding of how changes in the fundamental building blocks of chromatin can alter gene expression programs to drive oncogenic transformations. This paper by Nacev et al reports additional 'oncohistone' mutations in histone H3 at R2, R8, and R26. These mutations impact methylation of lysines important for gene regulation, notably H3K4 and H3K27. These affects apparently occur in cis, in the same nucleosomes that carry the mutant histone, but they impact large domains that are normally repressed by PRC2-mediated H3K27me3. The loss of repression in the H3R2 and R26 mutant bearing cells is associated with defective differentiation, consistent with impaired PRC2 functions.

The discovery of additional recurrent cancer associated missense mutations in histone H3 is interesting and impactful. In general, the experiments are well executed and the findings will be of broad interest.

We thank the reviewer for this gracious and positive feedback on our manuscript. We are grateful that the review agrees with us that this work is of broad interest. We have followed the reviewer's helpful suggestions to further improve the manuscript as detailed in our point-by-point responses below.

A few suggestions to make the paper even stronger:

1. The use of the term 'second generation' to describe these H3 mutations could cause confusion, as it could be taken to mean these mutations as subsequent to or evolve from the previously defined mutations. The term also downplays the importance of the current findings, suggesting they might be less important than the previously defined mutations. Why not just say the findings here expand the list of known mutations to additional sites, without referring to them as 'second' anything?

We thank the reviewer for this helpful comment and have updated the manuscript text to replace the term 'second generation' with language that better reflects our intended meaning.

2. Understanding the data in Fig 1 is critical to understanding the rest of the paper. As such, the text describing these experiments needs to be more clear. As written, it seems the blots shown are of cell lysates (and that is what is written in the figure legend), but if that is true, there is no way the authors could know if the changes in lysine methylation were occurring in cis with the R mutations. The references mentioned (8-13) and the methods indicate these are likely actually blots of affinity isolations of the tagged histones. Please add a sentence or two to the text to make this clear.

We assume that the reviewer intended to reference Figure 2 and not Figure 1 since the former includes the immunoblotting data. We agree these data are critical to the paper and should be described clearly to the reader. The revised manuscript now includes the following sentence in the figure legend: "The upper band represents the epitope-tagged mutant (or wildtype control) histone, whereas the lower band represents the endogenous wildtype histone pool". While we used an affinity purification to prepare material for the mass spectrometry experiments, the data in Figure 2 are generated from whole cell lysates, taking advantage of the differential migration of the epitope-tagged transgenic mutated (or WT control) histone versus the endogenous protein. We chose to analyze the whole cell lysates to simultaneously examine the *in trans* effects on the entire endogenous histone pool instead of only the heterotypic nucleosomes that would have been analyzed in an affinity purified sample. The same approach was used by others to describe the *in cis* effects of

a H3.3 G34 oncohistone mutant¹. We cited that study elsewhere in our manuscript, but in the revised version now also cite it along references the reviewer mentioned to stress that we are using an established approach. While some experiments in references 8 and 13 (original manuscript numbering) did use affinity purified histones, others used either whole cell lysates or acid extracted histones without enriching for the mutant. For example, Figure panels 2A and 2B in Lewis *et al.*² and Figure 2A in Lu *et al.*³ used whole cell extracts from 293T cells expressing an epitope tagged histone transgene to demonstrate marked global *in trans* effects on the bulk histone pool. In our experiments, we used a similar approach to show the absence of an effect on the global pool, but a marked effect on the transgenic histone mutants.

3. One explanation for the effects of the R2 mutant (and the R26 mutant) on PRC-mediated repression is a change in expression of components of the PRC complex itself. Please indicate whether any change in the RNA or protein levels of the PRC components are altered in the R mutants.

We thank the reviewer for raising this interesting hypothesis. To explore it, we analyzed the expression of the core PRC2 subunits (EED, SUZ12, EZH2, Rbbp4/6) in both 293T and MPCs. We have added this analysis as a new Supplementary Fig S2A in the revised manuscript and have updated the text to include a discussion of the results. While expression of core PRC2 subunits was reduced in the H3R2C mutant relative to H3WT and in MPCs, this was not the case in 293T cells suggesting changes in PRC2 subunit expression do not explain the effects on H3K27 methylation, which were seen in both cell lines. The expression of the H3R26C mutant did not affect PRC2 subunit expression in either cell line, which does not support a mechanism for H3K27me3 loss that depends upon changes in PRC2 subunit expression in that mutant either.

4. How do the effects of the R2 or R26 mutations in differentiation compare to the effects of PRC loss? This could be assessed through direct comparison in these cells, or at least discussed more fully in the Discussion.

We have followed the reviewer's excellent suggestion and have added text and references to the discussion in the revised manuscript to highlight similarities between the effects of the HR26C mutant and PRC2 loss of function on embryonic stem cell differentiation.

5. Both R2 and R26 are subject to methylation themselves, but Rme is not mentioned at all in the paper. Could loss of Rme be related to the effects here? Can the authors use artificial nucleosomes with R2me or R26me as substrates for PRC2? At the very least, Rme and PRMT functions should be discussed in the context of possible cross-talk with Kme and PRC (and MLL) functions.

The reviewer raises an excellent point about the crosstalk between arginine methylation and H3K4 and H3K27 methylation. We agree the manuscript would be improved by additional discussion of this topic and how it might relate to the phenotypes we observed in the presence of the H3R2 and R26 mutants. We have revised the manuscript accordingly. Specifically, we discuss that 1) there is well-characterized cross-talk between H3R2me2a and H3K4me3, which are anti-correlated and that H3R2me2 is enriched at inactive promoters⁴. We note that relationship would not explain why loss of H3R2 methylation (in the H3R2 mutant) leads to the observed loss of H3K4 methylation. 2) Similarly, H3R2me2s impairs PRC2 activity compared to unmethylated H3R2 substrates⁵, which would not explain loss of the PRC2 product upon expression of an H3R2 mutant in the three cell lines examined in our manuscript.

To experimentally explore whether these relationships held within our system, we tested whether arginine methyltransferase inhibition would phenocopy either the H3R2C or H3R26C mutant effects on H3K4me3 and H3K27me3 on the transgenic H3WT histone tail (new Supplementary Figure 3). Following a recently reported approach⁶, we used inhibitors of type I arginine methyltransferases (MS023)⁷, the predominant type II arginine methyltransferase, PRMT5 (GSK591)⁸, or both, to reduce methylarginine modifications. Type I and type II arginine methyltransferases catalyze arginine monomethylation, type I arginine methyltransferases catalyze asymmetric dimethyl arginine, and type II arginine methyltransferases catalyze symmetric dimethyl arginine⁹. We established doses of each drug that reduced monomethyl arginine as well as asymmetric and symmetric

dimethylarginine on multiple proteins using these PTM-specific antibodies. At these doses, there was no reduction in H3K4me3 or H3K27me3 on the transgenic H3WT tail. On the contrary, the combination of type I and II inhibitors slightly increased H3K4me3 and H3K27me3. This is in keeping with the expected crosstalk based on the literature cited above. Thus, potential loss of H3R2- and H3R26 methylation is unlikely to explain the observed phenotypes on H3K4me3 and H3K27me3.

A separate set of data also suggests that loss of H3R2 or R26 methylation is unlikely to be an important factor in explaining the H3K4 and H3K27 methylation changes. We further examined the comprehensive PTM analysis of the H3WT N-terminal tail from our middle-down mass spectrometry data to understand the prevalence of H3R2 and R26 methylation. We did not detect any H3R2 methylation and H3R26me1/2 was detected in <10% of H3WT population, which is consistent with prior studies¹⁰. While mass spectrometry is not as sensitive for arginine methylation as for lysine methylation, the loss of a minor population of Rme is unlikely to explain the near complete loss of the abundant H3K27 methyl marks observed by immunoblot. To make the mass spectrometry data more accessible to the reader, we now include a table of the comprehensive PTMs ratios from our middle-down mass spectrometry experiments (new Supplementary Table 2) to augment the access to the repository of mass spectrometry data we had provided with the original submission.

With respect to loss of H3K27me3 in the H3R26C mutant, the *in vitro* PRC2 methyltransferase experiments support a direct mechanism independent of H3R26 methylation status. PRC2 activity is nearly completely abolished in the R26 mutant relative to H3WT, which harbored an *unmodified* R26. While it is theoretically possible that introducing H3R26 methylation on a H3WT tail would enhance PRC2 activity well above that observed for the H3WT substrates with unmodified R26, this interesting but unlikely biochemical possibility could be explored in future work specifically focused on that question.

While we acknowledge that no experiment can perfectly recapitulate all potential crosstalk scenarios when considering the vast combinatorial nature of histone PTMs, the literature, the new experiments on using arginine methyltransferase inhibitors in MPCs, the mass spectrometry data in the 293T system, and the *in vitro* PRC2 assays suggest that loss of H3R methylation does not explain our observed phenotypes.

Reviewer #2 (Remarks to the Author): Expert in functional genomics and epigenomics, chromatin modifications, cancer genomics, and development

In this manuscript, Nacev and colleagues focused on the H3 R2/R26 oncohistones with mutation frequencies similar to those of classical at H3K27, G34, or K36. They demonstrated that H3 N-terminal R2 and R26 mutations impaired PRC2 activity *in vivo* and *in vitro*. They further examined H3R2C and R26C mutation-impacted H3K27me3 domains and downstream transcriptional pathways. By using mesenchymal progenitor cells and murine embryonic stem cell-derived teratomas models, they showed H3R26C mutant could affect lineage commitment of these progenitor cells.

Overall, this study expands the repertoire of oncohistones and presents novel findings on H3R2 linked to a remote site (H3K27). However, before its acceptance for publication, the following points should be addressed:

1. An *in vivo* cancer model should be used for the analysis of H3 R2/R26 oncohistones' contribution to carcinogenesis.

We agree with the reviewer that studies of the H3R2 and R26 histone mutations in cancer models would be of great interest to the field. For example, these models will be important in answering the questions posted in our discussion about whether different histone mutations that affect the same PTM are context dependent and if they potentiate the activity of additional drivers. We note that the study of oncohistones in animal models has been historically challenging, perhaps in part because of these nuances. For example, important work in an *in vivo* model of the *cis-acting* classical oncohistone mutations at H3G34 in giant cell of bone¹¹ was not published until almost a decade after these mutations were first reported despite intense interest. In addition, a recently published germline knock-in model of different H3G34 mutations¹² did not reveal a cancer phenotype,

suggesting the need to carefully model co-occurring mutations. Lastly, models for H3K27M, another classical oncohistone, are similarly challenging, with current state of the art cell-based models relying upon complex stem cell-derived systems^{13,14}. Given this collective experience in the oncohistone field, we respectfully suggest that creating a model to study the contribution of the H3R2 and R26 histones in carcinogenesis is beyond the scope of this initial disclosure and will require extensive effort beyond what is feasible in the 6-month timeframe allowed for revisions.

2. While arginine methylation at H3 R2 has been well established, and H3K4me3 is severely impaired in all three cell lines with H3R2C mutation (Figure 2A-C), the changes of H3K4me3 and H3R2me2 profiles and their connection with H3K27me3 and downstream pathways in H3R2C cells should be analyzed.

We thank the reviewer for these suggestions. We have now expanded the manuscript to include further analysis of H3K4me3 CUT&RUN data in the H3R2C mutant as suggested. This analysis revealed regions of peak overlap in which gain of H3K27me3 was associated with decreased H3K4me3 (enrichment score 10.39, $p < 1.38 \times 10^{-47}$), which is expected given that in general, H3K4me3 and H3K27me3 mark different chromatin states. Interestingly, we also observed a significant overlap of regions where H3K4me3 and H3K27me3 were both lost in the H3R2C mutant compared to wildtype (enrichment score 8.00, $p < 2.30 \times 10^{-58}$). There was no significant overlap of regions that lost H3K27me3 and gained H3K4me3 in H3R2C mutant expressing MPCs (enrichment score 1.08, $p < 0.35$). These results are consistent with a potential direct effect of the H3R2C mutant given that both H3K4me3 and H3K27me3 are depleted on the mutant histone as shown in Figure 2A-C. This contrasts with the association between gained H3K27me3 and lost H3K4me3, which is best explained by an indirect effect, perhaps through redistribution of H3K27me3. Notably, the genes associated with overlapping decreased H3K4me3 and H3K27me3 peaks in the H3R2C are enriched in GO terms associated with polycomb regulated genes, which again suggests a direct effect.

To address the reviewer's question regarding downstream pathways, we broadened the GO term enrichment analysis of the genes associated with the overlapping decreased H3K27me3 and H3K4me3 peaks. This analysis identified significant enrichment of pathways relevant to development including, "mesenchymal differentiation", "stem cell differentiation", and "muscle organ development". We have updated the revised manuscript with these findings including a new paragraph in the results section, a new Supplemental Figure 7, a new Supplemental Table 6.

Regarding the question of the relationship between H3R2me2 and H3K27me3, we kindly draw the reviewer's attention to our response to the fifth point raised by Reviewer 1, above, which addresses this topic in detail.

Reviewer #3 (Remarks to the Author): Expert in single-cell RNAseq, development, chromatin modifications, and computational genomics

Nacev et al. have reported findings that two tumor-associated H3 mutations (H3R2C and H3R26C) reduce H3K27me3 in multiple cell lines and disrupt differentiation as evidenced in ESC-derived teratomas and mesenchymal progenitor cells.

While the manuscript is generally well-composed, there are significant concerns regarding genomic data analysis.

Major:

1. In Figure 3B, C, and Supplementary Figure 3, the inferred decline in H3K27me3 signals isn't convincingly evidenced by the heatmaps, although the aggregated curves provide some clarity. It's advisable for the authors to consider scatter plots (with each dot representing one peak) that illustrate per-peak signals (reads per million) between WT and R2C/R26C. Subsequent statistical tests should be conducted to demonstrate the significance of this decrease.

We have followed the reviewer's helpful suggestion and generated scatter plots comparing the per peak counts per million (CMP) between WT and each of the mutant and present these data in a new Supplementary Figure

4D in the revised manuscript. We understood the reviewer's comment as a request to show all the peaks, not just the significantly changing differential peaks, so have generated the plot accordingly. These plots demonstrate lower H3K27me3 signal in the H3R2C and R26C mutants and mid-range and strongest peaks, with higher signal in the mutants at the weakest peaks. Taking the data altogether, we also now present boxplots representing CPM in each group in a new Supplementary Figure **4E** demonstrating an overall decrease in H3K27me3 signal in the H3R mutants compared to H3WT. Following the reviewer's suggestion, we perform a statistical test to demonstrate the significance of the decrease between the signal in the H3WT and arginine mutants within replicates. As shown on the in the figure, the decreased signal in the mutants is significant with the greatest p-value $< 1.0 \times 10^{-27}$.

Having added the analysis requested by the reviewer, we also highlight that Supplementary Figure **4A**, which we had included in the original manuscript, provides a volcano plot of the H3K27me3 signal in arginine mutant versus H3WT samples. This plot may be of particular interest to the reviewer since this demonstrates both the magnitude and significance of the signal change at specific peaks. As described in the main text, the volcano plot demonstrates both gained and lost peaks, which we determined are associated with known polycomb regulated genes. We also stress that the original Figure **3B** and **3C** referenced by the reviewer included peaks that have already be filtered to include only those that significantly change in the respective mutants based on the statistical tests incorporated into our analysis package. Notably, to be inclusive of all the data, we show all significantly changed peaks in those panels regardless of the magnitude of the change. Because of this, we also provided in the original Figure **3F** and **3G** heatmaps of the H3K27me3 differential peaks that were both statistically significant and had the greatest magnitude of loss in one or the other mutant. We hope that the reviewer will agree that these differences are clear. Importantly, the key finding we focus on in these data is that H3R2C and R26C mutants affect locus specific H3K27me3 domains and cause depletion of H3K27me3 at specific loci. This claim is supported by the above-mentioned analysis and with specific examples provided in the IGV tracks shown in Figure **3** and Supplementary Figure **6** (previous Supplementary Fig. **S4**).

2. The result in Figure 3A shows the overlap between the K27me3 differential peaks and Polycomb Regulated Gene Sets. The method to assign each peak to a gene is not clear. A more direct way is to compare these peaks with Polycomb-bound peaks and use another irrelevant protein's peaks as the control set.

We regret that that method for assigning peaks to a gene was not clear. We have now clarified the methods section to note that peaks were annotated to genes based on proximity; peaks that overlapped with TSS regions were considered for comparisons.

We thank the reviewer for suggesting an additional method to support the overlap of the H3K27me3 differential peaks with polycomb regulated gene sets. We have now provided an additional analysis to further support this conclusion. However, to avoid introducing any unintentional bias or assumption in the suggested selection of the "irrelevant" comparator, we instead performed a test based on random permutations. Specifically, we randomly selected genes from our global gene list to create a background list of genes equivalent to our observed significant genes. GO term enrichment analysis was then performed on these permuted background gene lists. This process was repeated 1000 times to generate the permutation test distribution, which was plotted and presented in a new Supplementary Figure **5** in the revised manuscript. As is demonstrated by the plots shown in that figure, there was no significant overlap with these permuted gene cohorts and the gene sets shown to significantly overlap with the H3K27me3 peaks shown in Figure **3A**. This supports our assertion that the statistically significant overlap between the H3K27me3 peaks and polycomb regulated gene sets is unlikely to have occurred at random.

3. The authors claimed that loss of H3K27me3 "occurred more often within 1 kb of promoters" using Figure 3H which doesn't show sample sizes. It may be worth plotting the H3K27me3 signals over the promoter regions and comparing them with the non-promoter peaks to show the difference. This can be coordinated with the first point, denoting promoters in a different color in the scatter plot.

We have now included the samples size for each group in a revised Figure **3H**.

In addition, following the reviewer's suggestion for the data visualization, we have also used a scatter plot to represent the per peak H3K27me3 signal in H3R2C and H3R26C compared to H3WT for those peaks where the signal is decreased in the mutant and the peak is either at promoters or in distal intergenic regions (**Figure R1**). We selected the latter since we referred to that specifically in the original text in reference to Figure 3H. Because the scatter plots are ordered by signal level, there is an added dimension of complexity in the visualization compared to the original plot in Figure 3H, which is based on the counts of differential peaks in each genomic region. In addition, the scatter plot visualization is limited in the number of different genomic regions that can be shown given the overlap of datapoints and the need to color code the genomic regions. In contrast, the stacked barplot provided in Figure 3H easily presents all genomic regions. In the original results section, we referenced Figure 3H only to compare the number of peaks lost at specific genomic features. We therefore suggest that the simpler visualization of the data in original Figure 3H would be clearer to the reader and have retained it in the manuscript. As noted, we have added sample sizes for each group per the reviewer's helpful suggestion.

Figure R1. Peak signal (CPM) in H3WT versus H3R mutants at promoters or distal intergenic regions. Only peaks lost in the mutant are plotted.

4. The Gene Ontology analysis is rather superficial and many leading terms are generic ones such as cell adhesion and rRNA processing. What are the driving genes underpinning the interesting GO terms the authors have highlighted? Some examples need to be provided.

To better allow the reader to assess the genes relating to the GO terms of interest, we have now added a new Supplementary Table 8, which includes the specific genes that contribute to the GO terms we highlighted in Figure 4A and 4B. In addition, we have now revised the main text to provide specific examples of genes from each of the pathways we had discussed in the original version of the manuscript in reference to Figure 4B. These genes demonstrate clear relevance to the pathways of interest. For example, we highlight *Bmp4* and *Sox9* in relation to the Osteoblast Differentiation pathway and *Cebpa* and *Pparg* in relation to the Regulation of Fat Cell Differentiation pathway.

5. The motif analysis is useful in understanding the pathways associated with H3K27me3 loss but only Arnt was mentioned. What other TFs have been called? The manuscript should elaborate on the top hits, either within the figure annotations or the main body.

The data for the Tomtom and DREME outputs were provided as referenced in the main text in the original Supplementary Table 6 (now Supplementary Table 9 in the revised manuscript). Using the analysis approach described in the methods, we identified several motifs for which the only significant association with a known binding motif was with ARNT. Therefore, we focused on the ARNT related hypoxia and angiogenesis pathways in the GSEA analysis in Figure 4E to highlight the potential functional relevance of the putative ARNT binding sites.

6. The single-cell RNA-seq data seem to conflict with the conclusion that R26C disrupts differentiation because in Figure 6D, the R26C 5-aza cells are more peripheral on the UMAP whereas WT 5-Aza is closer to the DMSO control region. The authors should draw arrows to point out the hypothetical differentiation path and explain why they think R26C disrupts differentiation.

We thank the reviewer for raising the important point and have taken the reviewer's excellent suggestion to better define a differentiation pathway in the single cell data. We carefully considered the reviewer's suggestion to "...draw arrows to point out the hypothetical differentiation path..." to represent our hypothesis of how this aberrant differentiation takes place. However, we determined that a more robust approach would be to apply a formalized trajectory analysis to our data, which we have presented in a new Figure 7. In summary, this analysis demonstrates that after receiving a differentiation signal, much of the H3R26C population becomes further distanced in pseudotime compared to other samples. With this new analysis, we were able to better assess our initially proposed hypothesis that "that H3R26C-expressing cells might aberrantly or only partially differentiate" to suggest that at least a subset of the H3R26C-expressing population adopts an aberrant differentiation trajectory. We have updated the results and discussions sections accordingly. We thank the reviewer for the helpful suggestion that led us to present this further characterization of the differentiation phenotype in the H3R26 mutant.

Turing to the reviewer's comment that "the single-cell RNA-seq data seem to conflict with the conclusion that R26C disrupts differentiation...", we believe this can be addressed with a semantic clarification. As the reviewer points out, we have used the word "disrupts" in our text to explain the effects of the H3R26C mutant. It appears that our use of disrupt may have conveyed a meaning closer to an 'arrest' in differentiation. However, we intended the meaning to be closer to 'perturb'. In fact, in our original manuscript, we present the hypothesis that "that H3R26C-expressing cells might aberrantly or only partially differentiate" at the outset of the results section reporting the sc-RNA-seq data. This hypothesis was based on the data from *in vitro* differentiation and in teratoma experiments (original Fig. 5) and the title of that results section is: "H3R26C-mediated chromatin disruption is associated with aberrant or partially differentiated states". To better express our meaning elsewhere in the text, we have replaced the word "disrupts" in the results section title "H3R26C mutations disrupt mesenchymal differentiation in MPC" with the word "perturb" in the revised manuscript.

7. The authors used the top 20 PCs for scRNA-seq analysis which might overlook more structures. What's the rationale for choosing that? How does it compare with using 50 PCs?

We selected 20 PCs based on the PC versus Standard Deviation plot (Figure R2), which demonstrated that 20 PCs capture most of the variation. Based on that plot, using 50 PCs instead of 20 PCs would not contribute additional significant PCs to the analysis since 20 PCs is already at the 'elbow' of the plot.

Figure R2. PC versus Standard Deviation

8. Figure 6C shows strong association between certain clusters and sample identities. Have the authors tried Seurat's batch correction to see if the association is from technical effects?

The samples analyzed to generate the original Figure 6C were collected from a controlled experiment using defined genotypes and control/treatment conditions. The samples were also sequenced on the same

sequencer, the same flow cell, and on the same lane. Thus, these samples were from the same batch. Therefore, we did not apply a batch correction in the Seurat package.

9. How do the findings from the single-cell data correlate with the earlier-mentioned bulk data? Can the differential genes identified in the bulk dataset be parsed into lineage-specific programs using single-cell data?

We thank the reviewer for raising this question and have undertaken the analysis as suggested. We do want to highlight that an important consideration in interpreting the comparison is that the bulk RNA-seq and the scRNA-seq were not performed on samples grown under similar conditions. The bulk RNA-seq analysis compared H3R mutant versus H3WT MPCs that were proliferating in the absence of any differentiation cues. In contrast, the scRNA-seq was performed on cells subjected to a differentiation protocol in order specifically test a hypothesis related to differentiation effects. As described in the original methods section, the differentiating cells were permitted to reach confluences and were cultured for four weeks in the same dish after differentiation induction (or control treatment) following an established protocol. Hence, even the transcriptomes of the vehicle control groups in the differentiation assay would be expected to be biologically distinct from the transcriptomes reflected in the bulk RNA-seq experiments. Therefore, this is a scenario quite different from comparing bulk and single cell transcriptomes generated from aliquots of the same tissue sample or culture, for example.

With these caveats in mind, we correlated the single cell data with the bulk data as suggested by the reviewer. We present an example of this analysis in a new Supplementary Fig. **S9D-G** in which we shown module scores and sample-specific feature plots based on a selection of significant GO terms from the bulk-RNA-seq analysis reported in the original Fig **4A**. As discussed in the revised manuscript, this analysis supports a correlation between the bulk RNA-seq and scRNA-seq datasets despite biologic differences between inputs to the two transcriptomic analyses.

Minor:

1. The x-axis of Figure 1a lacks clarity. What does the length of each bar mean?

We appreciate the feedback and have updated the Figure **1A** legend and x-axis label in our revised manuscript.

2. Figure 1a and 1b share a y-axis; aligning them at the same height could enhance visual coherence. Alternatively, merging them might be effective—wherein the x-axis of one panel is transposed to the y-axis and dot size indicates mutation count.

We thank the reviewer for raising this point and have now adjusted the range of the y-axis of Figure **1A** to match that of Figure **1B**. In remaking the figure, we noted that mutations in H3.5 were included in the original Figure **1A**, but not Figure **1B**, and are therefore not included in the revised Figure **1A**. This small adjustment does not affect the conclusions from Figure **1** that mutations at H3R2, R8, and R26 are common compared to classical oncohistones. In the revised Figure **1A**, we now also plot the amino acids which are not mutated for added clarity since these are shown in Figure **1B**.

3. Figure 2F isn't showing much information when the gene clusters are not annotated. It might be moved to Figure 4?

We thank the reviewer for this comment. The key conclusion we draw from Figure **2F** is that the genes with decreased expression in H3R26C cluster with genes with decreased expression in H3R2C. This suggests a similar underlying effect on chromatin-dependent gene regulatory networks that we then explore later in the manuscript. For that reason, we respectfully suggest that leaving the panel in its current place will facilitate the logical flow of the manuscript. As for the clustering, we dive deeper into analysis of specific genes and groups

of genes with differential expression in Figure 4 and link these changes to the chromatin environment as presented in Figure 3. We hope that we have explained our logic for the figure ordering and that since this was a minor comment from the reviewer, we will be permitted some leeway for preserving the placement of this panel.

4. Figure 2E can label the sample IDs for individual data points

We have added labels for the replicate number corresponding to each of the points on the revised Figure 2E. The genotypes for these points are already provided in the legend include as part of the panel.

5. Line 228's "H2K27me3" should be "H3K27me3"

Thank you for calling this typo to our attention. We have corrected it in the revised manuscript.

6. Supplementary Table 6's second tab lacks protein names.

We thank the reviewer for raising this question. On Supplementary Table 9 (original Supplementary Table 6), the second tab (labeled "Dreme_Results") represents de novo motifs detected by DREME. As described in the methods section of our original manuscript, we use DREME first to identify any de novo motifs that are enriched in our region of interest. The motifs identified and listed on the "Dreme_Results" tab, are not based on binding sites for known proteins. Instead, using TomTom, we then compare these motifs to database of known motifs and their associated proteins (these results are listed on the tab "Tom_Tom_Results"). For this reason, there should not be protein names directly associated with the DREME output.

7. For the single-cell analyses in Figure 6, it would be useful to include more intuitive names than cluster IDs, especially in Figure 6C. For example, Group 9 can be called CD34+ MPC etc.

We agree that providing intuitive names to the clusters would be helpful to the reader. However, we have also endeavored to avoid an over-annotation of the clusters that would risk an improper interpretation of the cell populations identified. As described in the original manuscript, "Unlike scRNA-seq of tissues from an intact organism where defined populations of cells are expected, we anticipated analyzing a mixture of undifferentiated progenitor cells, differentiated mesenchymal populations, and intermediate or partially differentiated cells. Therefore, we expected to observe only a few defined cell populations based on markers from mature mouse tissues". This was indeed the case and we concluded it was best to provide a descriptive annotation for most clusters as detailed in the text itself and in the original Figure 6A. We do highlight two clusters that are consistent with adipocytes and muscle cells, which are clearly labeled. The main theme highlighted from Figure 6, including Figure 6C, is that specific clusters are enriched for H3R26C mutant cells after 5-aza treatment, which is not dependent on the somewhat subjective naming of annotations. We hope the rationale for our approach resonates with the reviewer.

Reviewer #4 (Remarks to the Author): Expert in cancer epigenetics, CUT&RUN, and mass spectrometry

The CUT&RUN assays and data analysis were performed by the standards in the field.

We are grateful that the reviewer agrees that both the CUT&RUN assays and the data analysis in our paper meet the standards in the field.

There are only two minor points that the authors should include:

-the number of unique reads per C&R library

We thank the reviewer for raising this point. We wish to highlight that the number of reads for each sample was included in the original reporting summary.

-why did the authors choose for a 2 hrs/4°C-pA/G-MNase digestion rather than for 30 min/0°C.

We chose these conditions for the incubation based on the protocol provided by the commercial supplier of the pAG-MNase used in our experiments. Specifically, as described in the methods, we followed the CUTANA (Epiccypher) protocol v1.6. Although this protocol has been iteratively updated by the vendor, the current version available on their website (v2.1) continues to use a 2-hour MNase digestion at 4°C, thereby suggesting the robustness of this approach. Importantly, all samples, including our controls, were processed using the same reaction conditions.

References

- 1 Shi, L., Shi, J., Shi, X., Li, W. & Wen, H. Histone H3.3 G34 Mutations Alter Histone H3K36 and H3K27 Methylation In Cis. *J Mol Biol* **430**, 1562-1565 (2018). <https://doi.org/10.1016/j.jmb.2018.04.014>
- 2 Lewis, P. W. *et al.* Inhibition of PRC2 activity by a gain-of-function H3 mutation found in pediatric glioblastoma. *Science* **340**, 857-861 (2013). <https://doi.org/10.1126/science.1232245>
- 3 Lu, C. *et al.* Histone H3K36 mutations promote sarcomagenesis through altered histone methylation landscape. *Science* **352**, 844-849 (2016). <https://doi.org/10.1126/science.aac7272>
- 4 Guccione, E. *et al.* Methylation of histone H3R2 by PRMT6 and H3K4 by an MLL complex are mutually exclusive. *Nature* **449**, 933-937 (2007). <https://doi.org/10.1038/nature06166>
- 5 Liu, F. *et al.* PRMT5-mediated histone arginine methylation antagonizes transcriptional repression by polycomb complex PRC2. *Nucleic Acids Res* **48**, 2956-2968 (2020). <https://doi.org/10.1093/nar/gkaa065>
- 6 Maron, M. I. *et al.* Type I and II PRMTs inversely regulate post-transcriptional intron detention through Sm and CHTOP methylation. *Elife* **11** (2022). <https://doi.org/10.7554/eLife.72867>
- 7 Eram, M. S. *et al.* A Potent, Selective, and Cell-Active Inhibitor of Human Type I Protein Arginine Methyltransferases. *ACS Chemical Biology* **11**, 772-781 (2016). <https://doi.org/10.1021/acscchembio.5b00839>
- 8 Duncan, K. W. *et al.* Structure and Property Guided Design in the Identification of PRMT5 Tool Compound EPZ015666. *ACS Medicinal Chemistry Letters* **7**, 162-166 (2016). <https://doi.org/10.1021/acsmchemlett.5b00380>
- 9 Yang, Y. & Bedford, M. T. Protein arginine methyltransferases and cancer. *Nat Rev Cancer* **13**, 37-50 (2013). <https://doi.org/10.1038/nrc3409>
- 10 Schwämmle, V. *et al.* Systems Level Analysis of Histone H3 Post-translational Modifications (PTMs) Reveals Features of PTM Crosstalk in Chromatin Regulation. *Molecular & Cellular Proteomics* **15**, 2715-2729 (2016). <https://doi.org/10.1074/mcp.m115.054460>
- 11 Khazaei, S. *et al.* H3.3G34W promotes growth and impedes differentiation of osteoblast-like mesenchymal progenitors in Giant Cell Tumour of Bone. *Cancer Discov* (2020). <https://doi.org/10.1158/2159-8290.CD-20-0461>
- 12 Khazaei, S. *et al.* Single substitution in H3.3G34 alters DNMT3A recruitment to cause progressive neurodegeneration. *Cell* **186**, 1162-1178 e1120 (2023). <https://doi.org/10.1016/j.cell.2023.02.023>
- 13 Funato, K., Smith, R. C., Saito, Y. & Tabar, V. Dissecting the impact of regional identity and the oncogenic role of human-specific NOTCH2NL in an hESC model of H3.3G34R-mutant glioma. *Cell Stem Cell* **28**, 894-905.e897 (2021). <https://doi.org/10.1016/j.stem.2021.02.003>
- 14 Bressan, R. B. *et al.* Regional identity of human neural stem cells determines oncogenic responses to histone H3.3 mutants. *Cell Stem Cell* **28**, 877-893.e879 (2021). <https://doi.org/10.1016/j.stem.2021.01.016>

REVIEWER COMMENTS

Reviewer #1 (Remarks to the Author):

The authors have thoughtfully and thoroughly addressed all previous comments and suggestions.

Reviewer #2 (Remarks to the Author):

This manuscript has been substantially improved, and I have no further comment.

Reviewer #3 (Remarks to the Author):

The authors have made significant improvements to their manuscript. However, the motif analysis remains a major concern, particularly with only one hit identified. I suggest utilizing established TF-motif databases such as CIS-BP to identify enriched known motifs. This alternative approach (which is common practice) may enhance sensitivity and better showcase the datasets' potential.

Reviewer #3 (Remarks on code availability):

The Github codes should be organized into one single repo.

Reviewer #4 (Remarks to the Author):

In their revised version the authors addressed my minor concerns in a satisfactory manner.

Response to reviewers

Manuscript ID:

We are grateful to all the reviewers for their time and effort in providing constructive feedback during the revision process.

Reviewer #1 (Remarks to the Author):

The authors have thoughtfully and thoroughly addressed all previous comments and suggestions.

Reviewer #2 (Remarks to the Author):

This manuscript has been substantially improved, and I have no further comment.

Reviewer #3 (Remarks to the Author):

The authors have made significant improvements to their manuscript.

However, the motif analysis remains a major concern, particularly with only one hit identified. I suggest utilizing established TF-motif databases such as CIS-BP to identify enriched known motifs. This alternative approach (which is common practice) may enhance sensitivity and better showcase the datasets' potential.

We appreciate this feedback and have followed Reviewer 3's suggestion to replace our original motif analysis with one based on CIS-BP. We have revised the manuscript text, Figure 4D and 4E, and Supplementary Table 9 accordingly. As predicted by the reviewer, this new approach highlighted additional motifs reflective of TFs that putatively bind the motifs associated H3K27me3 loss. As discussed in the revised text, many of these TFs recognize E-box motifs and are from the bHLH family. Particularly relevant to the mesenchymal context, these motifs include consensus motifs for the muscle lineage transcription factors MYOD1 and MYOG. Following from this, we now show a GSEA analysis demonstrating an association between H3R26C expression and expression of myogenesis pathway genes. Although this new analysis no longer identifies ARNT as a hit, we note that ARNT binds a modified E-box motif, which highlights a similar theme between the initial and subsequent analysis. We thank the reviewer for the suggestion of using the CIS-BP database and agree this approach provides a more helpful and hypothesis generating dataset for the reader.

Reviewer #3 (Remarks on code availability):

The Github codes should be organized into one single repo.

We agree that a single resource would be helpful, but also respectfully suggest that since the analysis are independent, having them separate helpful to ensure accreditation and maintenance of the code. If this logic does not resonate with the Reviewer we are happy to reorganize the GitHub.

Reviewer #4 (Remarks to the Author):

In their revised version the authors addressed my minor concerns in a satisfactory manner.

REVIEWERS' COMMENTS

Reviewer #3 (Remarks to the Author):

The authors nicely addressed all my concerns.